# Bridging Lottery Ticket and Grokking: Understanding Grokking from Inner Structure of Networks

## Abstract

Grokking is the intriguing phenomenon of delayed generalization: networks initially memorize training data with perfect accuracy but poor generalization, then transition to a generalizing solution with continued training. While reasons for this delayed generalization, such as weight norms and sparsity, have been discussed, the influence of network structure, particularly the role of subnetworks, remains underexplored. In this work, we link the grokking phenomenon to the lottery ticket hypothesis to investigate the impact of inner network structures. We demonstrate that using lottery tickets obtained at the generalizing phase (termed 'grokking tickets') significantly reduces delayed generalization on various tasks, including multiple modular arithmetic, polynomial regression, sparse parity, and MNIST. Through a series of controlled experiments, our findings reveal that neither small weight norms nor sparsity alone account for the reduction of delayed generalization; instead, the presence of a good subnetwork structure is crucial. Analyzing the transition from memorization to generalization, we observe that rapid changes in subnetwork structures, measured by the Jaccard distance, correlate strongly with improvements in test accuracy. We further show that pruning techniques can accelerate the grokking process, transforming a memorizing network into a generalizing one without updating the weights. Finally, we confirm the emergence of periodic inner-structures, indicating that the model discovers internally good structures (generalizing structures) suited for the task.

## 1 Introduction

Understanding the mechanism of generalization is a central question in understanding the efficacy of neural networks. Recently, Power et al. (2022) unveiled the intriguing phenomenon of *delayed generalization (grokking)*; neural networks initially attain a *memorizing network* $C_{\text{mem}}$ with the perfect training accuracy but poor generalization, yet further training transitions the solution to a *generalizing network* $C_{\text{gen}}$. This phenomenon, which contradicts standard machine learning expectations, is being studied to answer the question: *what underlies the transition between memorization and generalization?* (Liu et al., 2022; 2023a)

Regarding the relationship between generalization and deep learning in general, it is well known that *structure of networks* significantly impacts generalization performance. For instance, image recognition performance has greatly improved by leveraging the structure of convolution (Krizhevsky et al., 2012). Moreover, as shown in Neyshabur (2020), incorporating $\beta$-Lasso regularization into fully connected MLPs facilitates the emergence of locality—resembling the structures in CNNs—leading to improved performance in image tasks. From a slightly different perspective, Frankle & Carbin (2019) proposed the lottery ticket hypothesis (LTH), which suggests that good subnetworks (good structure) help to achieve better performance with better sample efficiency (Zhang et al., 2021). Also, Ramanujan et al. (2020) shows that exploring structures alone can achieve performance comparable to weight updates, suggesting that good subnetworks are enough to achieve generalized performance.

While the importance of structure is well known in general, its connection to the phenomenon of grokking has not been investigated enough. Similar to our study, several prior works connect the grokking phenomenon to the property of networks, e.g., weight norm and sparsity of networks. For

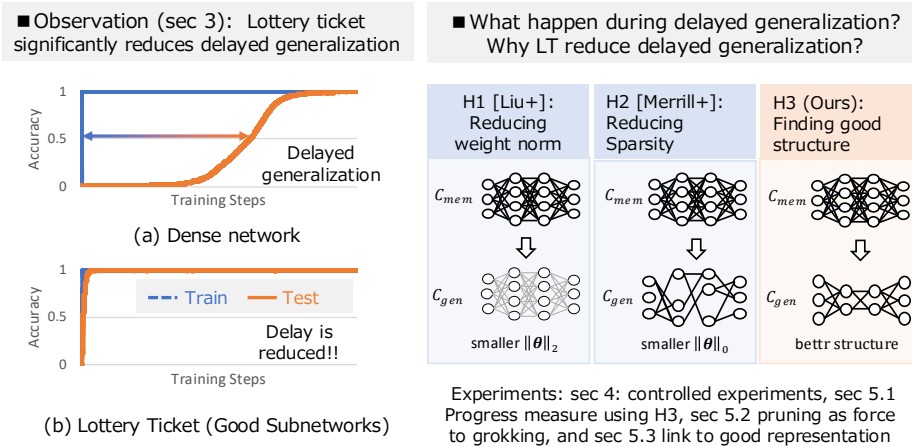

Figure 1: (Left) Accuracy of dense model and the lottery ticket obtained at generalizing solution (grokking ticket). When using a lottery ticket (good subnetworks), the train and test accuracy increase almost similarly, i.e., the *time from memorization ($t_{mem}$) to generalization ($t_{gen}$) has significantly accelerated*. Note that not only the subtraction ($t_{gen} - t_{mem}$) but the ratio ($t_{gen}/t_{mem}$) is also significantly improved, meaning that it's not just a matter of faster learning. (Right) Three hypotheses on why delayed generalization is reduced with a lottery ticket. We show that it is not due to a reduction in weight norm or an increase in sparsity, but rather the discovery of *good structure*.

example, Liu et al. (2023a) experimentally confirmed that generalizing solutions have smaller norms compared to memorizing solutions. The original paper (Power et al., 2022) showed that adding weight decay during training is necessary for triggering grokking. However, our work differs from these works by discussing the relationship between the process of discovering good structures (i.e., subnetworks) and grokking rather than merely reducing the weight norm.

To investigate the role of structure in the grokking phenomenon, we first demonstrate that when using the lottery ticket obtained at generalizing solution (referred to as grokking tickets), delayed generalization is significantly reduced. Figure 1 (left) illustrates that the train and test accuracy increase almost simultaneously with grokking tickets, unlike randomly initialized dense networks where delayed generalization occurs. As will be shown later, this result is related to the pruning rate, with proper pruning rates resulting in less delay. We conducted further experiments from several perspectives to understand why delayed generalization is significantly reduced with grokking tickets. First, as illustrated in Figure 1 (right), we decompose the differences between the grokking ticket and a randomly initialized dense network into three elements: (1) small weight norms, (2) sparsity and (3) good structure. We investigate which of these elements contribute to the significant reduction of delayed generalization. For (1) weight norms, we find that dense networks with the same initial weight norms as the grokking ticket do not generalize faster, indicating weight norm is *not* the cause. For (2) sparsity, networks with the same parameter size using well-known pruning methods (Wang et al., 2019; Lee et al., 2019; Tanaka et al., 2020) also do not generalize faster, indicating sparsity is *not* the cause. These results suggest that (3) good structure is essential for understanding grokking.

Subsequently, based on the above results, we analyze whether good structure exploration *truly* occurs during the transition phase. Using the jaccard distance (Paganini & Forde, 2020) as the metric of structural distance, we show that the structure of the subnetwork is rapidly changing during the transition between memorization and generalization. Furthermore, based on these results that structure exploration is crucial for generalization performance, we demonstrate that pruning facilitates generalization. We employ the edge-popup algorithm (Ramanujan et al., 2020), which finds a good structure while keeping the weights unchanged and demonstrates that the memorizing network can be transferred to the generalizing network through pruning without weight updates. Finally, we analyzed the specific network structure exhibited by the grokking ticket. In the task of Modular Addition, it is known that generalization performance improves when the network acquires periodic representations (Nanda et al., 2023; Liu et al., 2022; Pearce et al., 2023). By examining the weights of the grokking ticket, we observed that they also exhibit periodic representations, indicating that

the grokking ticket has acquired a periodic structure. This finding suggests that the model discovers internally good structures (generalizing structures) suited for the task.

In summary, our contributions are below:

- By linking lottery tickets and grokking, we first investigate the role of inner structure (subnetworks) in the grokking process. Our results show that using the lottery ticket significantly reduces the occurrence of delayed generalization.

- By decoupling the potential effect of lottery tickets into 1) weight norm, 2) sparsity, and 3) good structure and designing a set of controlled experiments (section 4), we further discuss why lottery tickets reduces the delayed generalization.

- We show that the change of inner structure (subnetworks obtained by the magnitude pruning) highly correlates with the change in test accuracy (section 5.1). We also show that pruning during training also accelerates the grokking process, and the memorizing solution can be transformed into a generalizing solution *with only pruning* (section 5.2).

- We show the grokking ticket has acquired a periodic structure, which implies that model has adapted to the task by learning a structure that facilitates generalization.

## 2 BACKGROUND

**Grokking** is a phenomenon where generalization happens long after overfitting the training data (as shown in Figure 1 (left)). The phenomenon was initially observed in the modular addition task ($(a + b) \mod p$ for $a, b \in (0, \cdots, p - 1)$), and the same phenomenon has been observed in more complex datasets, encompassing modular arithmetic (Gromov, 2023; Davies et al., 2023; Rubin et al., 2023; Stander et al., 2023; Furuta et al., 2024), semantic analysis (Liu et al., 2023a), n-k parity (Merrill et al., 2023), polynomial regression (Kumar et al., 2023), hierarchical task (Murty et al., 2023) and image classification (Liu et al., 2023a; Radhakrishnan et al., 2022). This paper mainly focuses on grokking in the modular arithmetic tasks commonly used in prior studies.

To understand grokking, previous works proposed possible explanations, including the slingshot mechanism (Thilak et al., 2022), random walk among minimizers (Millidge, 2022), formulation of good representation (Liu et al., 2022), the scale of weight norm (Liu et al., 2023a; Varma et al., 2023), simplicity of Fourier features (Nanda et al., 2023) and sparsity of generalizing network (Miller et al., 2023). Among those, one of the dominant explanations regarding how the network changes during the process of grokking are the simplicity of the generalization solution, particularly focusing on the weight norms of network parameters $\|\boldsymbol{\theta}\|_2$. For example, the original paper (Power et al., 2022) posited that weight decay plays a pivotal role in grokking, i.e., test accuracy will not increase without weight decay. Liu et al. (2023a) analyzed the loss landscapes of train and test dataset, verifying that grokking occurs by entering the generalization zone defined by L2 norm, with models having large initial values $\boldsymbol{\theta}_0$. More recently, Varma et al. (2023) demonstrated that the generalization solution could produce higher logits with smaller weight norms. In this paper, we examine the changes in the network's structure and demonstrate that the network is not simply decreasing its overall weight norms but searching for good structures within itself.

Several studies have proposed that acquiring good representations is the key to understanding grokking. For example, Power et al. (2022); Liu et al. (2022) explained that the topology of the ideal embeddings tends to be circles or cylinders within the context of modular addition tasks. Nanda et al. (2023) identified the trigonometric algorithm by which the networks solve modular addition after grokking and showed that it grows smoothly over training. Gromov (2023) showed an analytic solution for the representations when learning modular addition with MLP. Zhong et al. (2023) show, using modular addition as a prototypical problem, that algorithm discovery in neural networks is sometimes more complex. These studies support the quality of representation as key to distinguishing memorizing and generalizing networks; however, these studies do not explain what is happening within the network's structure.

The **lottery ticket hypothesis** proposed by Frankle & Carbin (2019) has garnered attention as an explanation for why over-parameterized neural networks exhibit generalization capabilities (Allen-Zhu et al., 2019). Informally, the lottery ticket hypothesis states that randomly initialized over-parameterized networks include sparse subnetworks that reach good performance after train, and

the existence of the subnetworks is key to achieving good generalization in deep neural networks. This claim was initially demonstrated experimentally, but theoretical foundations have also been established (Frankle et al., 2020; Sakamoto & Sato, 2022). More formally, the process involves the following steps:

1. Initialize dense network $f_{\boldsymbol{\theta_0}}$ and train the network for $t$ epochs to obtain the weights $\boldsymbol{\theta_t}$

2. Perform $k\%$ pruning on the trained network based on absolute values $|\boldsymbol{\theta_t}|$. This process, known as magnitude pruning, yields a mask $\boldsymbol{m}_t^k \in \{0, 1\}^{|\boldsymbol{\theta_t}|}$.

3. Reset the weights of the network to their initial values. $\boldsymbol{\theta_0}$ and get a subnetwork $f_{\boldsymbol{\theta_0} \odot \boldsymbol{m}_t^k}$, representing *lottery ticket*. Train the subnetwork for $t'$ epochs and obtain $f_{\boldsymbol{\theta_{t'}} \odot \boldsymbol{m}_t^k}$.

After the discovery of the lottery tickets, Ramanujan et al. (2020) show that there exist **strong lottery tickets**, which achieve good performance without weight update. They use the **edge-popup** algorithm (Ramanujan et al., 2020) to selects subnetworks based on a score $\boldsymbol{s}$ ($\boldsymbol{s} \in \mathbb{R}^{|\boldsymbol{\theta_0}|}$). In other words, when pruning a certain proportion $k$ of weights from the given weights $\boldsymbol{\theta_0}$, the model predicts using edges with the top $(1 - k)$ scores in a forward pass. For a detailed description of edge-popup, refer to Appendix H. In section 5, we use the edge-popup algorithm to check if pruning can be worked as a force to accelerate the grokking process.

## 3 LOTTERY TICKETS SIGNIFICANTLY REDUCES DELAYED GENERALIZATION

### 3.1 EXPERIMENT SETUP

Following (Power et al., 2022) and other grokking literatures (Nakkiran et al., 2019; Liu et al., 2022; Gromov, 2023; Liu et al., 2023a), we constructed a dataset of equations of the form: $(a + b)\%p = c$. The task involves predicting $c$ given a pair of $a$ and $b$. Our setup uses the following detailed configurations: $p = 67, 0 \le a, b, c < p$. The dataset size is 2211, considering all possible pairs where $a \ge b$. We split it into training (40%) and test (60%) following (Liu et al., 2022).

Following Liu et al. (2022), we design the MLP as follows. Firstly, we map the one-hot encoding of $\boldsymbol{a}, \boldsymbol{b}$ with the embedding weights $W_{emb}$: $\boldsymbol{E}_a = W_{\text{emb}}\boldsymbol{a}, \boldsymbol{E}_b = W_{\text{emb}}\boldsymbol{b}$. We then feed the embeddings $\boldsymbol{E}_a$ and $\boldsymbol{E}_b$ into the MLP as follows:

$$\text{softmax}(\sigma((\boldsymbol{E}_a + \boldsymbol{E}_b)W_{\text{in}})W_{\text{out}}W_{\text{unemb}}) \tag{1}$$

where $W_{\text{emb}}, W_{\text{in}}, W_{\text{out}}$, and $W_{\text{unemb}}$ are the trainable parameters, and $\sigma$ is an activation function ReLU (Nair & Hinton, 2010). The dimension of the embedding space is 500, and $W_{\text{in}}$ projects into 48-dimensional neurons. Following (Nanda et al., 2023), we used the AdamW optimizer (Loshchilov & Hutter, 2019) with a learning rate $10^{-3}$, the weighting of weight decay $\alpha = 1.0, \beta_1 = 0.9$, and $\beta_2 = 0.98$. We initialize weights as $\boldsymbol{\theta_0} \sim \mathcal{N}(0, \kappa/\sqrt{d_{\text{in}}})$, where $d_{\text{in}}$ represents the dimensionality of the layer preceding each weight. If nothing is specified, assume $\kappa = 1$. Let us assume we have training datasets $\mathbf{S}_{\text{train}}$ and test datasets $\mathbf{S}_{\text{test}}$, and train a neural network $f(\boldsymbol{x}; \boldsymbol{\theta})$ where $\boldsymbol{x}$ is an input and $\boldsymbol{\theta}$ represents weight parameters of the networks. Specifically, the network is trained using AdamW over a cross-entropy loss and weight decay (L2 norm of weights $\|\boldsymbol{\theta}\|_2$):

$$\underset{\boldsymbol{\theta}}{\arg\min} \, \mathbb{E}_{(\boldsymbol{x},y)\sim\boldsymbol{S}} \left[ \mathcal{L}(f(\boldsymbol{x}; \boldsymbol{\theta}), y) + \frac{\alpha}{2}\|\boldsymbol{\theta}\|_2 \right].$$

To quantitatively measure how much-delayed generalization is reduced, we define $t_{\text{mem}}$ as the step at which the training accuracy exceeds $P\%$, and $t_{\text{gen}}$ as the step at which the test accuracy exceeds $P\%$. Following Kumar et al. (2024), we use $P = 95$ for modular arithmetic tasks. We use the proposition ($\tau_{\text{grok}} = t_{\text{gen}}/t_{\text{mem}}$) to measure the acceleration.

We compared the performance of 1) dense networks $f_{\boldsymbol{\theta_{t'}}}$ and 2) trained lottery tickets $f_{\boldsymbol{\theta_{t'}} \odot \boldsymbol{m}_t^k}$, where $t'$ is a training epoch to get the final score, $t$ is timing of pruning, and $k$ is a pruning ratio. As a special case, when $t \ge t_{\text{gen}}$, we denote the subnetworks as **grokking tickets**. We tested various $t$ and $k$ and investigated how they change the generalization speed. By default, we used $k = 0.6$.

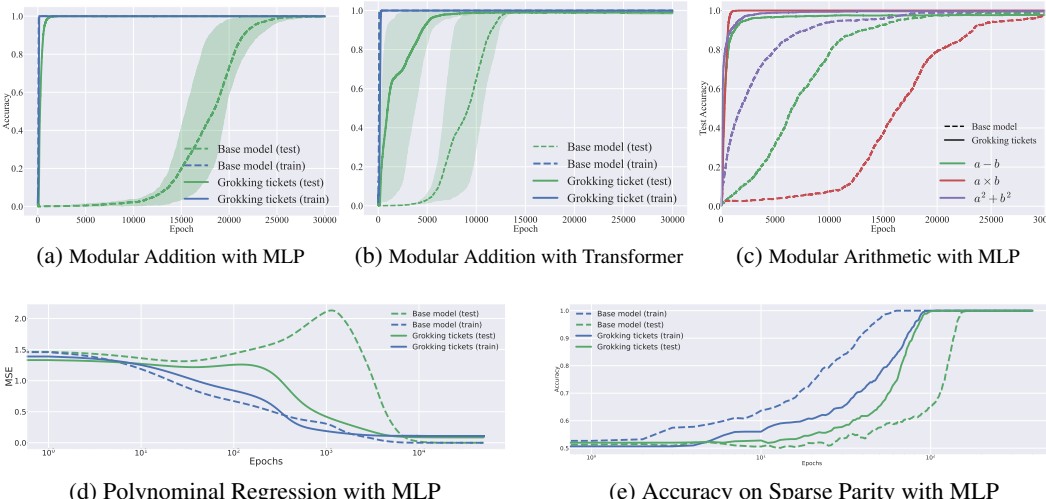

(a) Modular Addition with MLP  (b) Modular Addition with Transformer  (c) Modular Arithmetic with MLP

(d) Polynominal Regression with MLP  (e) Accuracy on Sparse Parity with MLP

Figure 2: Comparing the grokking speed of dense networks and grokking tickets on various setups. (a) Modular addition with MLP, (b) Modular addition with Transformer, and (c) Other modular arithmetic tasks (represented by color) and experiments other than modular arithmetic: (d) loss on polynomial regression, (e) accuracy on sparse parity. The dashed line represents the accuracy of the base model, and the solid line represents that of grokking tickets. In all setups, the time to generalization ($t_{gen}$) is reduced by grokking tickets.

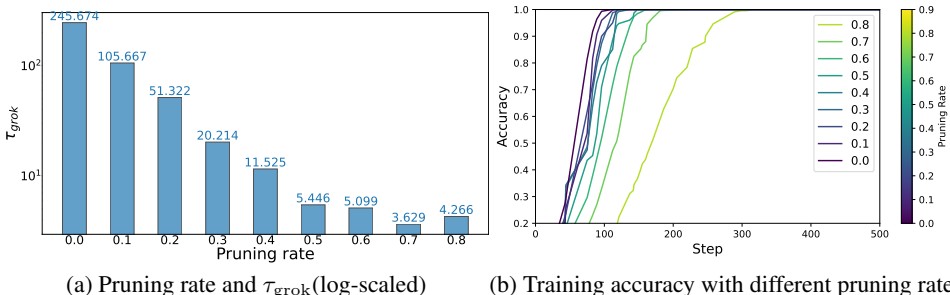

(a) Pruning rate and $\tau_{\text{grok}}$(log-scaled)  (b) Training accuracy with different pruning rate

Figure 3: Quantitative comparison of grokking speed among different pruning rates. Note that pruning rate = 0.0 corresponds to the dense network. The definition of the $\tau_{grok}$ is explained in subsection 3.1

## 3.2 RESULTS

Figure 2-(a) shows the test accuracy of the grokking ticket and the base model on the modular addition task. The base model refers to a dense model trained from random initial values. The grokking ticket shows an improvement in test accuracy at nearly the same time as the improvement in training accuracy of the base model. In Figure 2-(b), using experiments with transformers, the result also shows that grokking tickets result in less delay of generalization. Following Power et al. (2022), we conducted experiments on various modular arithmetic tasks to demonstrate the elimination of delayed generalization. Figure 2-(c) shows a comparison of the base model (dashed line) and grokking ticket (solid line) with various modular arithmetic tasks. Moreover, following Kumar et al. (2023); Pearce et al. (2023), we also demonstrate that delayed generalization is reduced by the grokking ticket in both the polynomial regression and sparse parity tasks in Figure 2-(d,e). The results show that grokking tickets significantly reduced delayed generalization even on various tasks. See Appendix B for a detailed explanation of the experimental setup.

Figure 3-(a) quantify the relationship between pruning rate and $\tau_{\text{grok}} = t_{\text{gen}}/t_{\text{mem}}$ (log-scaled). When the pruning rate is 0.7, $\tau_{\text{grok}}$ reaches its minimum, indicating that grokking ticket significantly reduce delayed generalization.

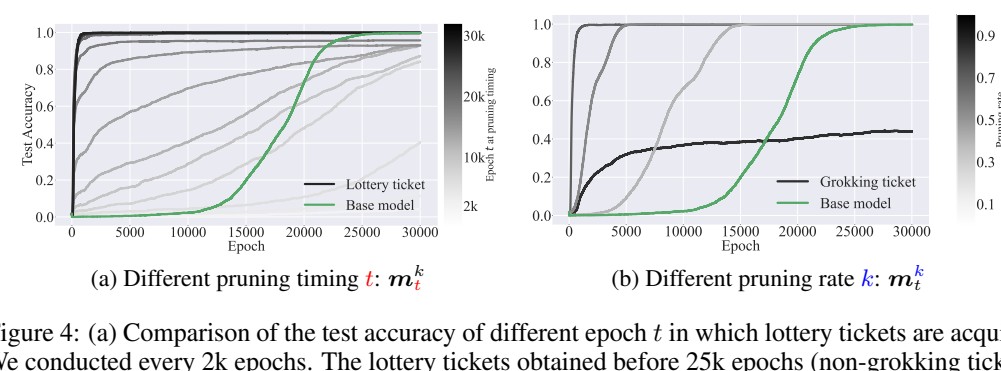

(a) Different pruning timing $t$: $\boldsymbol{m}_t^k$          (b) Different pruning rate $k$: $\boldsymbol{m}_t^k$

Figure 4: (a) Comparison of the test accuracy of different epoch $t$ in which lottery tickets are acquired. We conducted every 2k epochs. The lottery tickets obtained before 25k epochs (non-grokking tickets) do not fully generalize. Additionally, this generalization ability corresponds to the test accuracy of the base model. The lottery tickets obtained after 25k epochs (grokking tickets) reduced delayed generalization. (b) The effect of pruning rate $k$ on grokking tickets. We conducted every 0.2 pruning rate. Most pruning ratios (0.1, 0.3, 0.5, and 0.7) accelerate the generalization, indicating that the above observation does not depend heavily on the selection of the pruning ratio.

Note that, as shown in Figure 3-(b), the $t_{\text{mem}}$ is *delayed* when using a higher pruning rate, meaning that *grokking tickets the grokking ticket are not simply accelerating the entire learning process but specifically speeding up the transition from memorization to generalization*.

In Figure 4-(a), we compare the test accuracy of different epoch $t$ in which lottery tickets are acquired. The results show that lottery tickets obtained before 25k epochs (non-grokking tickets) do not fully generalize. Additionally, this generalization ability corresponds to the test accuracy of the base model. On the other hand, lottery tickets obtained after 25k epochs (grokking tickets) get perfect generalization and reduce delayed generalization. In Figure 4-(b), we investigate the effects of pruning rate in grokking ticket, indicating if it's too large (e.g., 0.9), it can't generalize; if it's too small (e.g., 0.3), it doesn't generalize quickly enough.

## 4 DECOUPLING LOTTERY TICKETS: NORM, SPARSITY, AND STRUCTURE

Figure 2 indicates that grokking ticket facilitates generalization. To better understand the inner workings, we decouple the potential benefits of lottery tickets into (1) small weight norms, (2) higher sparsity, and (3) good structure, as shown in Figure 1 (right). We conduct a series of controlled experiments using a grokking ticket to decouple each effect. Firstly, we find that dense networks with the same initial weight norms as the grokking ticket do *not* generalize faster. Additionally, compared to networks with the same sparsity as grokking tickets, we show mere sparsity also does *not* promote generalization. These results indicate that the reason why lottery tickets significantly accelerate the generalization process is better attributed to the **structure of networks**, rather than small weight norm and sparsity.

### 4.1 CONTROLLING WEIGHT NORM OF INITIAL NETWORK

Liu et al. (2023a); Varma et al. (2023) stated that the norm of a network's weights is a key factor for grokking. Liu et al. (2023a) assert that grokking occurs norm of the weights enter the 'Generalized zone' (Fort & Scherlis, 2018) through regularization. To investigate whether the weight norm or the good structure are more plausible explanations of delayed generalization, we prepared two dense models with the same L2 and L1 norms as the grokking ticket, named the 'controlled dense model.' Such dense models are obtained through the following process:

1. Obtain lottery tickets after full generalization $\boldsymbol{m}_{t_{\text{gen}}}^k$.

2. Get weight $L_p$ norm ratio $r_p = \frac{\|\boldsymbol{\theta}_0 \odot \boldsymbol{m}_{t_{\text{gen}}}^k\|_p}{\|\boldsymbol{\theta}_0\|_p}$

3. Create weights $\boldsymbol{\theta}_0 \cdot r_p$ with the same $L_p$ norm as the grokking ticket.

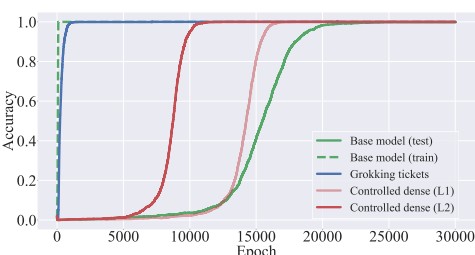 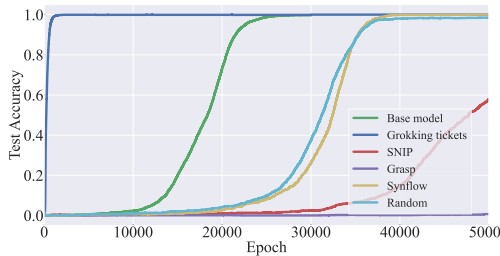

(a) Controlling initial weight norms (L1 and L2)    (b) Controlling sparsity w/ pruning at initialization

Figure 5: (a) Test accuracy dynamics of the base model, grokking ticket, and controlled dense model (L1 norm and L2 norm). The grokking ticket reaches generalization much faster than other models. (b) Comparing test accuracy of the different pruning methods. All PaI methods perform worse than the base model or, in some cases, perform worse than the random pruning. These results indicate neither the weight norm *nor* the sparsity alone is the cause of delayed generalization.

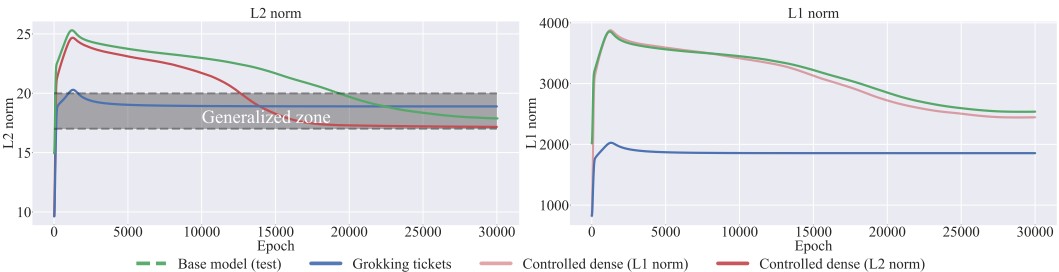

Figure 6: (Left) L2 norm dynamics of the base model, grokking ticket, and controlled dense model. (Right) L1 norm dynamics of the base model, grokking ticket, and controlled dense models. From the perspective of the L2 norm, all models appear to converge to a similar solution (Generalized zone). However, from the perspective of L1 norms, they converge to different values.

Figure 5-(a) shows the test accuracy of the base model, grokking ticket, and controlled dense models. Despite having the same initial weight norms, the grokking ticket arrives at generalization much faster than both controlled dense models. This result indicates that the delayed increase in test accuracy is attributable not to the weight norms but to the discovery of good structure. The left of the Figure 6 shows the dynamics of the L2 norms for each model. Similar to Liu et al. (2023b), L2 norms decrease in correspondence with the rise in test accuracy, converging towards a 'Generalized zone'. However, as shown on the right side of the Figure 6, the final convergence points of the L1 norms vary for each model. This phenomenon of having similar L2 norms but smaller L1 norms suggests that good subnetwork (grokking ticket) weights become stronger, as indicated in Miller et al. (2023). Similar results have also been observed in Transformer (Figure 15). These results demonstrate that the weight norm itself is insufficient to explain grokking.

## 4.2 Controlling Sparsity

In subsection 4.1, we show the discovery of good structure (sparse network) is a more critical factor than weight norm in explaining the delayed generalization. However, it raises questions about whether sparse models are crucial for grokking or grokking tickets possess good properties beyond mere sparsity. In this section, using various pruning methods, we show that mere sparsity is also insufficient for the explanation of delayed generalization. We compared the grokking ticket with subnetworks that had the same level of sparsity but were identified using different pruning methods. Specifically, we tested three well-known pruning at initialization (PaI) methods (Grasp (Wang et al., 2020), SNIP (Lee et al., 2019), and Synflow (Tanaka et al., 2020)) and random pruning as baseline methods. For details on each of the pruning methods, refer to the section Appendix G. Figure 5-(b) compare the transition of the test accuracy of these PaI methods and the grokking ticket. The results show that all PaI methods perform worse than the base model or, in some cases, perform worse than

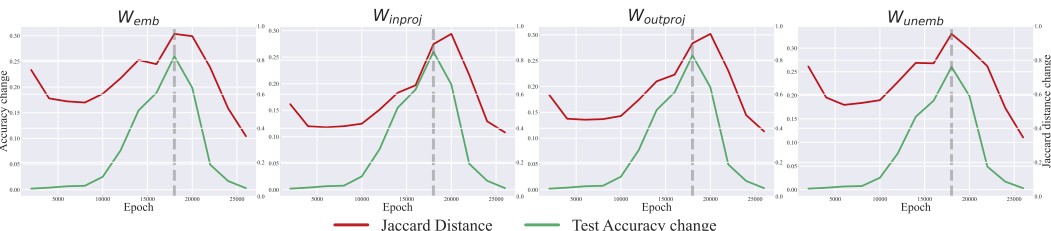

Figure 7: Comparison of jaccard distance (red) and changes in test accuracy (green) on each layer. The jaccard distance is represented as $\mathrm{JD}(t + \delta t, t)$, and the test accuracy change is the difference between epoch $t + \delta t$ and $t$. The vertical line marks the most drastic change in test accuracy. When there is a significant change in test accuracy, the jaccard distance (structural change) increases rapidly.

the random pruning. The results show that poor selection of the subnetworks hurts the generalization, suggesting that the grokking ticket holds good properties beyond just a sparsity.

## 5 UNDERSTANDING GROKKING FROM INNER STRUCTURE OF NETWORKS

In section 4, we demonstrate that weight norm and sparsity are insufficient as an explanation for delayed generalization and suggest that the reason why lottery tickets significantly reduce delayed generalization can be better attributed to the structure of subnetworks. Questions are: (1) How is the good property of subnetworks acquired during training? How does it correspond to the mystery of the delayed generalization? (2) Can we accelerate grokking using pruning during traning? (3) How does the inner structure perspective correspond to the good representations discussed in prior work?

### 5.1 PROGRESS MEASURE: STRUCTURAL SHIFT CAPTURE THE GENERALIZATION TIMING

In this section, we conduct a more rigorous analysis of how the good structure is acquired, and we show that the discovery of the good structure corresponds to an improvement in test accuracy. Firstly, we propose a metric of structural changes in the network, named *jaccard Distance* (JD) using approach Paganini & Forde (2020); Jaccard (1901). We measure the jaccard distance between the mask at epoch $t + \delta t$ and $t$.

$$\mathrm{JD}(\boldsymbol{m}_{t+\delta t}, \boldsymbol{m}_t) = 1 - \frac{|\boldsymbol{m}_{t+\delta t} \cap \boldsymbol{m}_t|}{|\boldsymbol{m}_{t+\delta t} \cup \boldsymbol{m}_t|}$$

$\boldsymbol{m_t}$ represents a mask obtained at $t$ epoch via magnitude pruning and $\delta t$ is 2k epoch. If the two structures differ, this metric is close to 1 and vice versa. Test accuracy change is also represented as a difference between test accuracy at epoch $t + \delta t$ and $t$. In Figure 7, the red line represents the results of the changes in test accuracy and jaccard distance between the mask at epoch $t + \delta t$ and $t$ on each layer. The results show that during significant changes in test accuracy (16k-20k), the maximum change in the mask corresponds, indicating that the discovery of good structure corresponds to an improvement in test accuracy. In the Appendix C, we demonstrate that similar results are obtained for both the polynomial regression and sparse parity tasks.

### 5.2 PRUNING DURING TRAINING: PRUNING PROMOTE GENERALIZATION

Based on the result that the discovery of a good structure corresponds to an improvement in test accuracy, we demonstrate that pruning alone can transition from memorizing solutions to generalizing solutions *without weight update*, and furthermore, the combination of pruning and weight decay promotes generalization more effectively than mere regularization of weight norms.

To verify this, we introduce edge-popup (Ramanujan et al., 2020), a method that learns how to prune weights without weight updates. In edge-popup, each weight is assigned a score, and these scores are updated through backpropagation to determine which weights to prune. For details regarding edge-popup, refer to the Appendix H. We validate our claim by optimizing using three different methods.

**WD** : Training from $\boldsymbol{\theta}_{\mathrm{mem}}$ using **Weight Decay** *with* weight update (same as base model).

Table 1: Test accuracy changes with different optimization methods starting from memorized solutions. WD (Weight Decay) reflects the regularization effect of weight decay, using the same optimization as the base model. In EP w/o WD (Edge-Popup without Weight Decay), accuracy improves solely through pruning, without weight updates. Combining pruning with weight decay in EP w/ WD results in faster generalization than weight decay alone.

| Epoch | 600 | 1000 | 1400 | 2000 |
|---|---|---|---|---|
| WD | $0.53 \pm 0.31$ | $0.95 \pm 0.03$ | $1.00 \pm 0.00$ | $1.00 \pm 0.00$ |
| EP w/o WD | $0.68 \pm 0.19$ | $0.80 \pm 0.17$ | $0.84 \pm 0.16$ | $0.92 \pm 0.06$ |
| EP w/ WD | $\mathbf{0.82 \pm 0.04}$ | $\mathbf{0.96 \pm 0.01}$ | $0.99 \pm 0.00$ | $1.00 \pm 0.00$ |

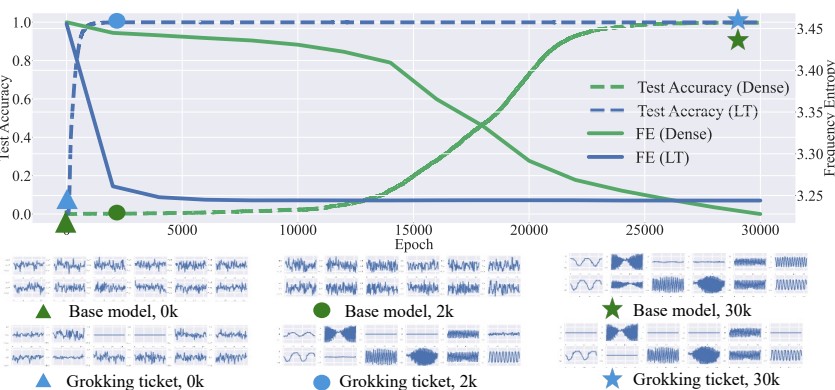

Figure 8: (top) Frequency entropy (FE) and test accuracy of the base model and the grokking ticket. The grokking ticket converges to a smaller frequency entropy much faster than the base model. (bottom) The transition of the input-side weights for each neuron of the base model and grokking ticket. The marks correspond to the epochs of the marks in FE dynamics. The results indicate that grokking tickets acquire good structures for generalization as periodic structures.

**EP w/o WD** : Training from $\theta_{\mathrm{mem}}$ using **Edge-Popup** *without* weight update.

**EP w/ WD** : Training from $\theta_{\mathrm{mem}}$ using **Edge-Popup** and **Weight Decay** *with* weight update.

In Table 1, the result in EP w/o WD shows the network exhibits a generalizing performance (0.92) without any change in weights (*merely by pruning weights*). Additionally, the EP w/ WD result shows the fastest improvement in test accuracy and is the most effective in promoting generalization. These insights suggest that practitioners may improve generalization by incorporating methods that directly optimize beneficial structures rather than solely relying on traditional regularization techniques like weight decay. Our findings highlight the potential of grokking tickets to inform the development of new, structure-oriented regularization techniques.

### 5.3 ACQUISITION OF GOOD STRUCTURES AS PERIODIC STRUCTURES

The previous study (Pearce et al., 2023; Nanda et al., 2023) shows that models acquire **periodic representations** when generalizing in Modular Addition (see details in Appendix L). In this section, following prior research, we examine what kind of structure is acquired by grokking tickets. Our analysis of the network weights after generalization reveals a clear periodicity (see the Base model at **30k** steps in Figure 8 (bottom)). Notably, grokking tickets develop this periodic structure much earlier than the base model (see the Grokking ticket at **2k** steps in Figure 8 (bottom)). This finding highlights that grokking tickets acquire a task-adaptive structure. To quantify this periodicity as a good structure, we introduce Fourier Entropy (FE) as follows. In general, the discrete Fourier transform $\mathcal{F}(\omega)$ of a function $f(x)$ is defined as follows:

$$\mathcal{F}(\omega) = \sum_{x=0}^{N} f(x) \exp\left(-i\frac{2\pi\omega x}{N}\right)$$

In this case, since we want to know the periodicity of each neuron's weights, $f(x)$ is the weight of the $i$-th neuron on the $j$-th input, and $d$ is the dimension of the input. Then, the Fourier Entropy is calculated as follows:

$$FE = -\sum_{i=1}^{n} p_i \log p_i$$

Here, $p_i$ is the normalized value of $\mathcal{F}(\omega)$, and $n$ represents a number of neurons. A low value of FE indicates that the **frequency of the weights of each neuron has little variation, converging to specific frequencies**, which shows that the network has acquired task-adaptive structure.

Figure 8 (top) shows the FE of the base model (green) and grokking ticket (blue). The results show that the grokking ticket has neurons with periodic structure at an early stage (2k epochs) and exhibits a rapid decrease in FE in the early epoch. This indicates that the model discovers internally good structures (generalizing structures) suited for the task (Modular Addition). To provide a more detailed analysis, we have added visualizations of the weight matrices and grokking ticket masks in Appendix K.

## 6 DISCUSSION AND RELATED WORKS

In this paper, we conducted a set of experiments to understand the mechanism of grokking (delayed generalization). Below is a summary of observations. (1) In subsection 3.2, the use of the lottery ticket significantly reduces delayed generalization. (2) In section 4, good structure is a more important factor in explaining grokking than the weight norm and sparsity by comparing it with the same weight norm and sparsity level. (3) In Figure 7, good structure is gradually discovered, corresponding to improvement of test accuracy. (4) In Table 1, pruning without updating weights from a memorizing solution increases test accuracy. (5) In Figure 8, finding a good structure corresponds to the acquisition of representations. Our work also contributes to a deeper understanding of grokking and may inspire a more rigorous approach to deep learning.

**Weight norm reduction** Liu et al. (2023a) suggests that the reduction of weight norms is crucial for generalization. In Figure 5, our results go further to show that, rather than simply reducing weight norms, the network discovers good structure (subnetwork), resulting in the reduction of weight norms.

**Representation learning** Liu et al. (2022); Nanda et al. (2023); Liu et al. (2023a); Gromov (2023) showed the quality of representation as key to distinguishing memorizing and generalizing networks. Figure 8 demonstrates that good structure contributes to the acquisition of good representation, suggesting the importance of inner structure (network topology) in achieving good representations

**Sparsity and Efficiency** Merrill et al. (2023) argued that the grokking phase transition corresponds to the emergence of a sparse subnetwork that dominates model predictions. While they empirically studied parse parity tasks where sparsity is evident, we are conducting tasks (modular arithmetic, MNIST) commonly used in grokking research and architecture (MLP, Transformer). Furthermore, Figure 5, we demonstrate not only sparsity but also that good structure is crucial.

**Regularization** Weight decay (Rumelhart et al., 1986) is one of the most commonly used regularization techniques and is known to be a critical factor in grokking (Power et al., 2022; Liu et al., 2023a). In Appendix E, we show if the good structure is discovered, the network generalizes without weight decay, indicating that weight decay works as a structure exploration, which suggests that weight decay contributes not to reducing weight but to exploring good structure. In addition, we tested pruning as a new force to induce generalization (Table 1).

**Lottery ticket hypothesis** The lottery ticket hypothesis (Frankle & Carbin, 2019) suggests that good structures are crucial for generalization, but it remains unclear how these structures are acquired during training and how they correspond to the network's representations. To the best of my knowledge, our paper is the first to connect grokking and the lottery ticket hypothesis, demonstrating how good structures emerge (subsection 5.1) and contribute to effective representations (subsection 5.3). Building on this, we further show that while faster generalization is a well-documented property of winning tickets, our work goes beyond by exploring how the discovery of good structures.

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

## A EXPERIMENTS WITH DIFFERENT SEEDS

To ensure reproducibility, we conduct experiments with three different seeds and present the results for experiments in Figure 2. In addition to accuracy, we plot the loss of train and test in Figure 9.

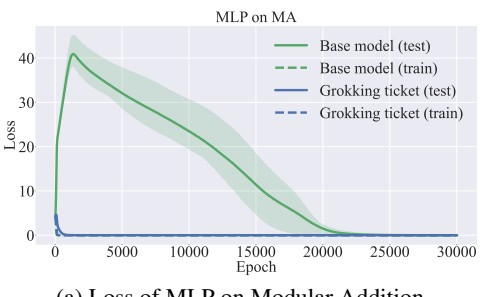

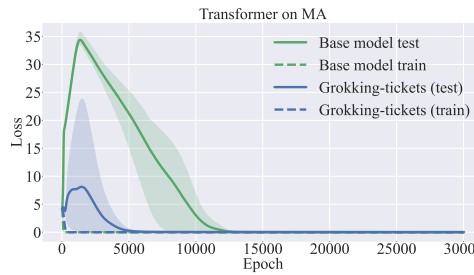

(a) Loss of MLP on Modular Addition

(b) Loss of Transformer on Modular Addition

Figure 9: Train and Test loss of base model and grokking ticket on MLP and Transformer. In both of architectures, grokking ticket convergent 0 faster than base model

Additionally, to ensure reproducibility, we conduct experiments with three different seeds and present the results for experiments in subsection 4.1 and Transformer architecture experiment.

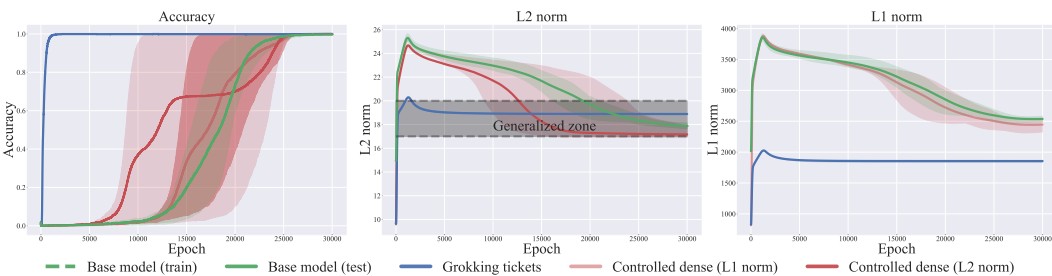

Figure 10: (Left) Test accuracy dynamics of the base model, grokking ticket, and controlled dense model (L1 norm and L2 norm) with three different seeds. The grokking ticket reaches generalization much faster than other models. (Center) L2 norm dynamics of the base model, grokking ticket, and controlled dense model. (Right) L1 norm dynamics of the base model, grokking ticket, and controlled dense models. From the perspective of the L2 norm, all models appear to converge to a similar solution (Generalized zone). However, from the perspective of L1 norms, they converge to different values.

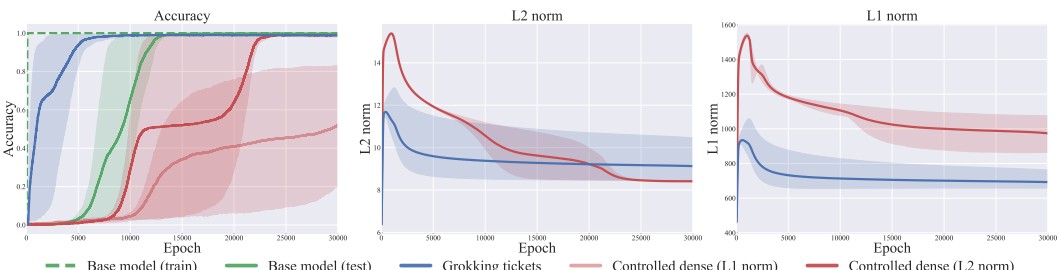

Figure 11: Test accuracy dynamics of the base model, grokking ticket, and controlled dense model (L1 norm and L2 norm) with three different seeds in the Transformer. The grokking ticket reaches generalization much faster than other models. (Center) L2 norm dynamics of the base model, grokking ticket, and controlled dense model. (Right) L1 norm dynamics of the base model, grokking ticket, and controlled dense models. From the perspective of the L2 norm, all models appear to converge to a similar solution (Generalized zone). However, from the perspective of L1 norms, they converge to different values.

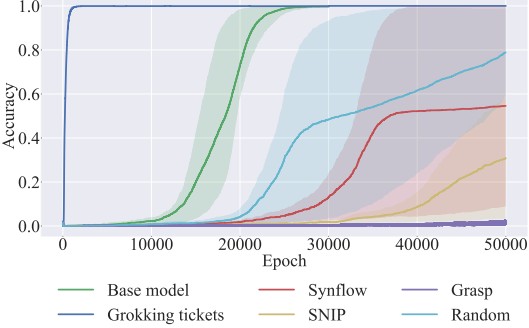

Figure 12: Comparing test accuracy of the different pruning methods. All PaI methods perform worse than the base model or, in some cases, perform worse than the random pruning with three different seeds.

## B    DIFFERENT CONFIGURATIONS OF THE TASK AND THE ARCHITECTURE.

### B.1    MLP FOR MNIST

We use 4-layer MLP for the MNIST classification. The difference from the regular classification is that we are using Mean Squared Error (MSE) for the loss. We adopted this setting following prior research (Liu et al., 2023a). In (Liu et al., 2023a), it was confirmed in the Appendix that grokking occurred without any problems, even when trying with cross-entropy. Figure 13 shows the test and train accuracy on various configurations. It is evident that grokking tickets accelerate generalization in all configurations, and the exploration of grokking tickets contributes to generalization.

### B.2    TRANSFORMER FOR MODULAR ADDITION

Similar to Nanda et al. (2023), we use a 1-layer transformer in all experiments. We use single-head attention and omit layer norm.

- **Hyperparameters:**
  - $d_{\text{vocab}} = 67$: Size of the input and output spaces (same as $p$).
  - $d_{\text{emb}} = 500$: Embedding size.
  - $d_{\text{mlp}} = 128$: Width of the MLP layer.
- **Parameters:**
  - $W_E$: Embedding layer.
  - $W_{\text{pos}}$: Positional embedding.
  - $W_Q$: Query matrix.
  - $W_K$: Key matrix.
  - $W_V$: Value matrix.
  - $W_O$: Attention output.
  - $W_{\text{in}}, b_{\text{in}}$: Weights and bias of the first layer of the MLP.
  - $W_{\text{out}}, b_{\text{out}}$: Weights and bias of the second layer of the MLP.
  - $W_U$: Unembedding layer.

We describe the process of obtaining the logits for the single-layer model. Note that the loss is only calculated from the logits on the final token. Let $x_i^{(l)}$ denote the token at position $i$ in layer $l$. Here, $i$ is 0 or 1, as the number of input tokens is 2, and $x_i^{(0)}$ is a one-hot vector. We denote the attention scores as $A$ and the triangular matrix with negative infinite elements as $M$, which is used for causal attention.

The logits are calculated via the following equations:

$$x_i^{(1)} = W_E x_i^{(0)} + W_{\text{pos}} x_i^{(0)},$$
$$A = \text{softmax}(x^{(1)\text{T}} W_K^{\text{T}} W_Q x^{(1)} - M),$$
$$x^{(2)} = W_O W_V (x^{(1)} A) + x^{(1)},$$
$$x^{(3)} = W_{\text{out}} \text{ReLU}(W_{\text{in}} x^{(2)} + b_{\text{in}}) + b_{\text{out}} + x^{(2)},$$
$$\text{logits} = \text{softmax}(W_U x^{(3)}).$$

Figure 13 shows the training and test accuracy of the base model (green) and the Grokking ticket (blue). Across datasets (e.g., Modular Addition, MNIST) and architectures (Transformer), the Grokking ticket (blue) consistently reaches generalization faster than the base model (green). These results underscore the importance of structural elements in grokking, regardless of the task or architecture.

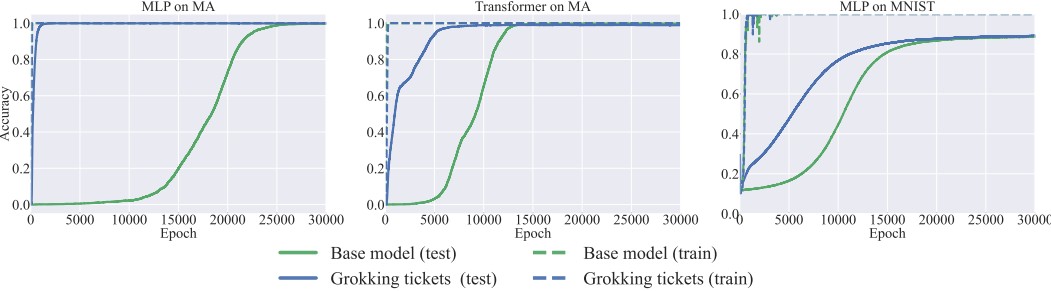

Figure 13: Comparison of base model (green) and Grokking ticket (blue). Each column corresponds to the different configurations of the task (Modular Addition and MNIST) and the architecture (MLP and Transformer). The dashed line represents the results of the training data.

# C  STRUCTURAL CHANGES IN TASKS OTHER THAN THE MODULAR ADDITION TASK

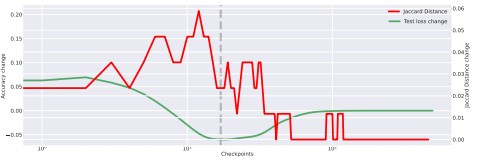

(a) Jaccard Distance on Polynominal Regression

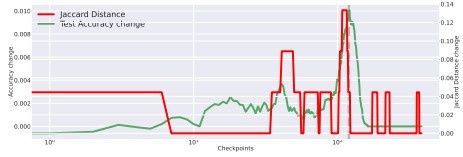

(b) Jaccard Distance on Sparse Parity

Figure 14: Jaccard distance change and test accuracy change on polynominal regression (a) and sparse parity (b). Structural changes (Jaccard distance) correspond to the acquisition of generalization ability.

# D  IS WEIGHT NORM SUFFICIENT TO EXPLAIN GROKKING IN TRANSFORMER?

Figure 15 show the accuracy of base model, Grokking ticket and Controlled dense (L1 and L2) on Transformer. The results show Grokking ticket generalize faster than any other model. The results suggests that even in the case of Transformer, the discovery of Grokking ticket is more important than weight norms.

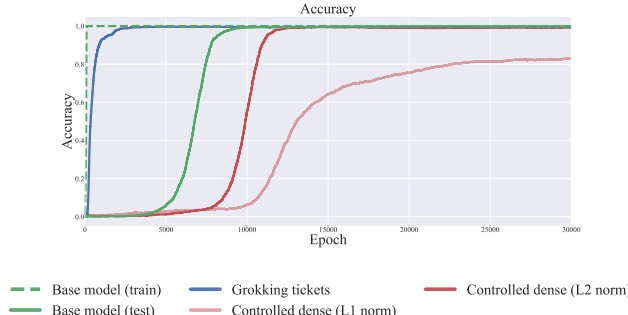

Figure 15: Accuracy of base model, Grokking ticket and Controlled dense on Transformer.

# E    WEIGHT DECAY WORK AS STRUCTURE EXPLORER

Our result is that the discovery of good structure happens between memorization and generalization, which indicates weight decay is essential for uncovering good structure but becomes redundant after their discovery.

In this section, we first explore the critical pruning ratio, which is the maximum pruning rate that can induce generalization without weight decay Figure 16-(a). We recognize that the critical pruning rate is between 0.8 and 0.9 because if the pruning rate increases to 0.9, the test accuracy dramatically decreases. Thus, we gradually increased the pruning rate in increments of 0.01 from 0.8 and found that the $k = 0.81$ is the critical pruning ratio. We then compare the behavior of the grokking ticket without weight decay and the base model. Figure 16-(b) shows the results of the experiments. The results show that if good structure is discovered, the network fully generalizes without weight decay, indicating that weight decay works as a structure explorer. In this section, we show that with precise

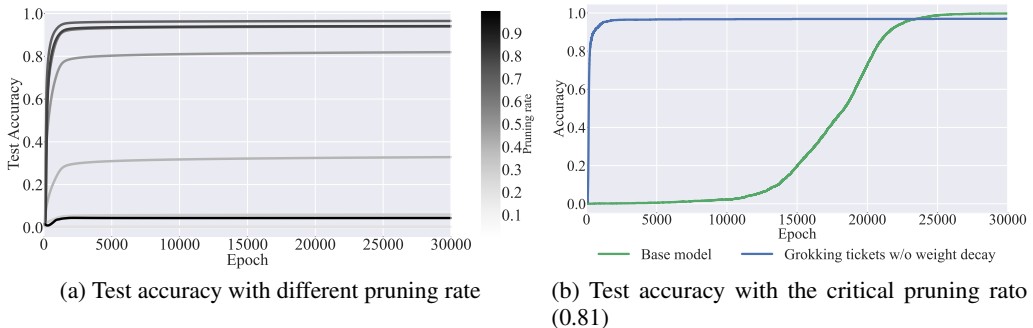

(a) Test accuracy with different pruning rate

(b) Test accuracy with the critical pruning rato (0.81)

Figure 16: The effect of pruning rate on test accuracy without weight decay(left). Test accuracy of grokking ticket with critical pruning rate (0.81) without weight decay(right).

pruning ratios, the grokking ticket does not require weight decay to generalize, indicating that weight decay is essential for uncovering good subnetworks but becomes redundant after their discovery. We first explore the *critical pruning ratio*, which is the maximum pruning rate that can induce grokking (Figure 16-a). In this case (Figure 16-a), we recognize that the critical pruning rate is between 0.8 and 0.9 because if the pruning rate increases to 0.9, the test accuracy dramatically decreases. Thus, we gradually increased the pruning rate in increments of 0.01 from 0.8 and found that the $k = 0.81$ is the critical pruning ratio. We then compare the behavior of the grokking ticket without weight decay ($\alpha = 0.0$) and the base model. Figure 16-b show the results of the experiments. As shown in the figure, the test accuracy reaches perfect generalization without weight decay. The results show that the grokking ticket with the critical pruning ratio does not require any weight decay during the optimization.

## F  DISCRETE FOURIER TRANSFORM

The discrete Fourier transform of $f(x)$ is as follows:

$$F(\omega) = \sum_{x=0}^{N-1} f(x) \exp(-i\frac{2\pi\omega x}{N})$$

The number of sample points ($N$) is 67, the same as the input dimension. We conduct this discrete Fourier transform for both the input-side weights and the output-side weights of each neuron. The Fourier entropy is as follows:

$$\text{FE} = -\frac{1}{K} \sum_{i=0}^{K} \sum_{\omega=0}^{[\frac{p}{2}]} F_i(\omega) \log F_i(\omega)$$

The number of neurons ($K$) is 48, the number of frequency points ($[\frac{p}{2}]$) is 33 and $p_i(t)$ is the normalized value at the $t^{th}$ sample point for the $i^{th}$ neuron. Figure 17 shows the transition of the input-side weights for all neurons of the base model and grokking ticket. The horizontal axis represents the input dimension (67), while the vertical axis represents weight values. The grokking ticket acquires representations at the early phase. Figure 18 shows that Fourier transform ($F(\omega)$

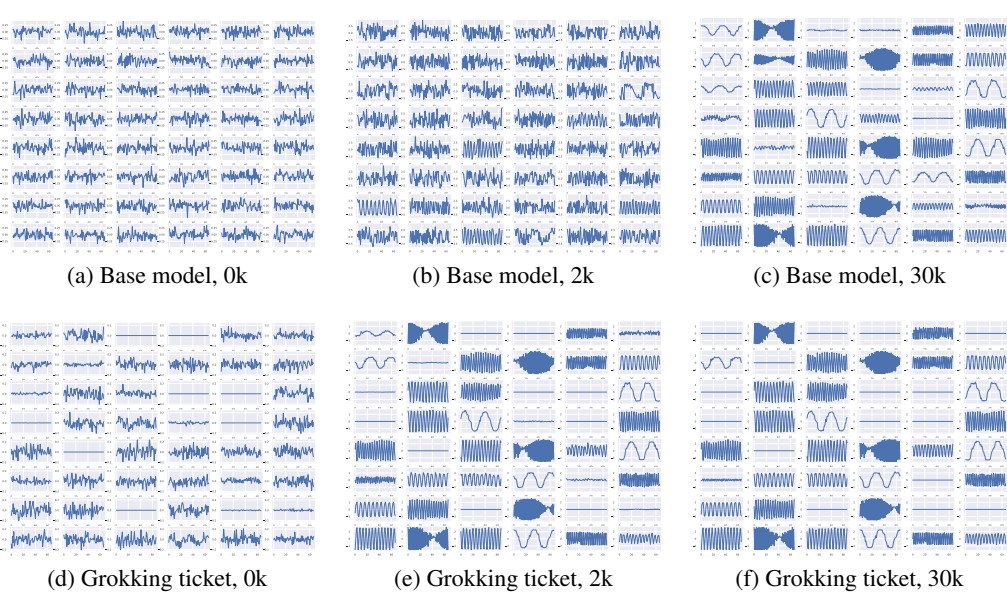

| (a) Base model, 0k | (b) Base model, 2k | (c) Base model, 30k |

| (d) Grokking ticket, 0k | (e) Grokking ticket, 2k | (f) Grokking ticket, 30k |

Figure 17: Comparison of the base model and grokking ticket. The top/bottom row shows the transition of the input-side weights for each neuron of the base model/grokking ticket. The horizontal axis represents the input dimension (67), while the vertical axis represents weight values. The grokking ticket acquires representations at the early phase.

of the inside-weights ($W_{in}$) of each neurons. After generalization, the frequency characteristics of most neurons in the base model are prominent for a specific frequency. In grokking ticket, neurons responsive to specific frequencies emerge at an early stage. (2k epochs).

## G  PRUNING AT INITIALIZATION METHODS

Currently, the methodologies of pruning neural networks (NN) at initialization (such as SNIP, GraSP, SynFlow) still exhibit a gap when compared to methods that use post-training information for pruning (like Lottery Ticket). Nonetheless, this area is experiencing a surge in research activity.

The basic flow of the pruning at initialization is as follows:

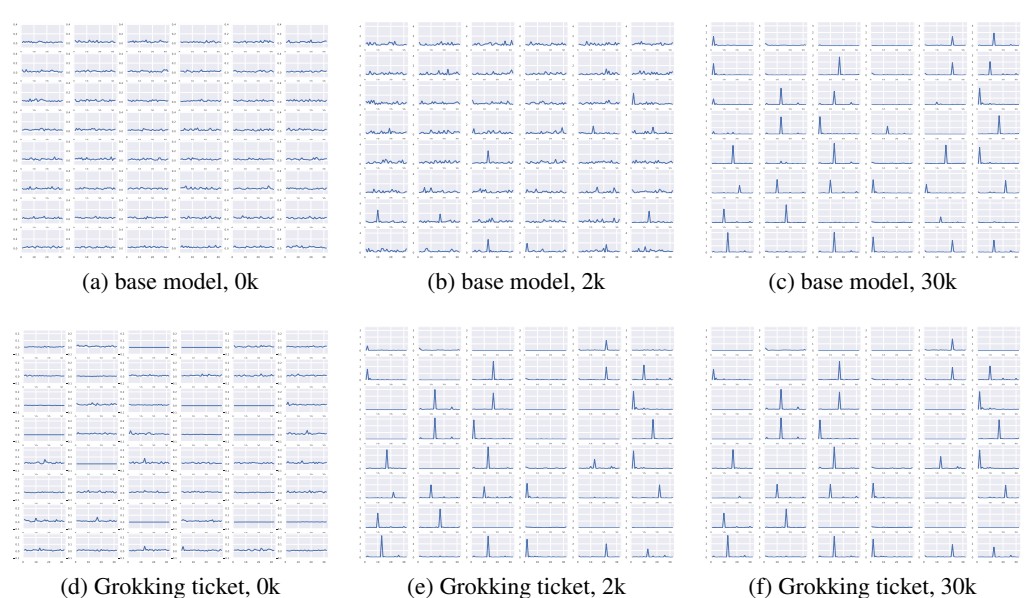

(a) base model, 0k       (b) base model, 2k       (c) base model, 30k

(d) Grokking ticket, 0k     (e) Grokking ticket, 2k     (f) Grokking ticket, 30k

Figure 18: Comparison of base model and Grokking ticket. The top/bottom row shows the transition of the frequency decomposition of weight values for each neurons of base model/Grokking ticket. The horizontal axis represents frequency, while the vertical axis represents amplification.

1. Randomly initialize a neural network $f(\boldsymbol{x}; \boldsymbol{\theta}_0)$.

2. Prune $p\%$ of the parameters in $\boldsymbol{\theta}_0$ according to the scores $S(\boldsymbol{\theta})$, creating a mask m .

3. Train the network from $\boldsymbol{\theta}_0 \odot \boldsymbol{m}$.

According to Tanaka et al. (2020), research on pruning at initialization boils down to the methodology of determining the score in the above process 2, which can be uniformly described as follows:

$$S(\boldsymbol{\theta}) = \frac{\partial R}{\partial \boldsymbol{\theta}} \odot \boldsymbol{\theta}$$

When the $R$ is the training loss $L$, the resulting synaptic saliency metric is equivalent to $|\frac{\partial L}{\partial \boldsymbol{\theta}} \odot \boldsymbol{\theta}|$ used in SNIP (Lee et al., 2019). $-(H\frac{\partial L}{\partial \boldsymbol{\theta}}) \odot \boldsymbol{\theta}$ use in Grasp (Wang et al., 2020).Tanaka et al. (2020) proposed synflow algorithm $R_{SF} = 1^T (\prod_{l=1}^{L} |\boldsymbol{\theta}^{|l|}|)1$. In section 4.2, all initial values were experimented with using the same weights and the same pruning rate.

## H    EDGE-POPUP ALGORITHM

We provide a detailed explanation of the edge-popup algorithm. Edge-popup **?** is a method for finding effective subnetworks within randomly initialized neural networks without weight updates.

**Basic Flow:**

1. Random Initialization: Initialize the weights $\mathbf{w}$ of a large neural network randomly.

2. Assign Scores: Assign a score $s_{ij}$ to each edge $w_{ij}$ randomly.

3. Select Subnetwork: Form a subnetwork $\mathcal{G}$ using only the edges with top $k\%$ scores.

4. Optimize Scores: Update the scores $s_{ij}$ based on the performance of the subnetwork $\mathcal{G}$.

**Score Updates:** Update the score $s_{ij}$ of each edge $w_{ij}$ using the following formula:

$$s_{ij} \leftarrow s_{ij} - \alpha \frac{\partial L}{\partial I_j} Z_i w_{ij}$$

- $\alpha$ is the learning rate
- $\frac{\partial L}{\partial I_j}$ is the gradient of the loss with respect to the input of the $j$-th neuron
- $Z_i$ is the output of the $i$-th neuron

**Actual Computation:** Forward Pass: Compute using only the edges whose scores $s_{ij}$ are in the top $k\%$.

$$I_j = \sum_{i \in V} w_{ij} Z_i h(s_{ij})$$

Here, $h(s_{ij})$ is 1 if $s_{ij}$ is in the top $k\%$ and 0 otherwise.

## I    LIMITATION

First, our study uses the framework of the lottery ticket hypothesis, so neural network models are within its scope. Therefore, grokking phenomena in non-neural networks, as seen in studies Levi et al. (2023); Miller et al. (2023), are outside the scope and are considered future work.

Additionally, although we have verified various experimental settings (Modular Arithmetic, MNIST classification, Polynomial regression, Sparse parity), the tasks and networks used are still simple, as in previous grokking studies. The lottery ticket hypothesis has been studied in more practical cases, so experiments in more practical scenarios are interesting and remain as future work.

In addition, our paper shows that structural changes correspond to generalization ability, but it does not propose specific metrics for what structural characteristics lead to good representations. Research on the network characteristics of good structures remains in future work.

# J GROKKING TICKETS IN NLP TASKS

Following Liu et al. (2023a), we conducted a sentiment analysis task on the IMDb dataset (Maas et al., 2011), which consists of 50,000 movie reviews classified as either positive or negative. The data was pre-processed by extracting the 1,000 most frequent words and tokenizing each review into an array of token indices, with less frequent words ignored and each review padded to a length of 500. For classification, we utilized a two-layer LSTM model (Hochreiter, 1997) with an embedding dimension of 64 and a hidden dimension of 128. The model was trained using the Adam optimizer with a learning rate of 0.001 and weight decay of 0.001 to minimize the binary cross-entropy loss, and 25% of the dataset was reserved for testing.

In our sentiment analysis task, which involves two-class classification with a chance rate of 50%, we observe distinct learning dynamics between the base model and the grokking ticket. The base model rapidly achieves 100% training accuracy; however, its test accuracy remains low until approximately 10,000 optimization steps, after which it begins to improve. On the other hand, the grokking ticket demonstrates a different behavior, with test accuracy improving almost simultaneously with training accuracy from the very beginning of the optimization process. These results suggest that the delayed generalization observed in grokking is closely linked to the discovery of optimal structural representations, highlighting its potential for uncovering meaningful patterns in the data.

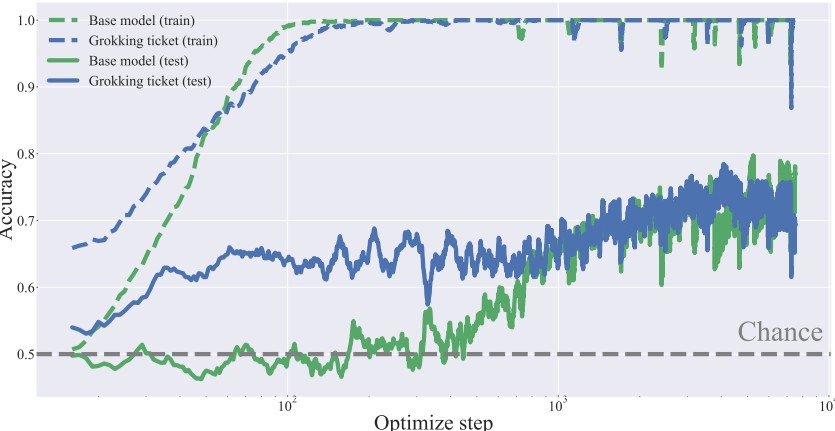

Figure 19: Results of sentiment analysis for two-class classification, with a chance rate of 50%. The base model (**green**) rapidly achieves 100% training accuracy, but test accuracy starts to improve only around 10k steps. In contrast, the grokking ticket (**blue**) shows simultaneous improvement in both training and test accuracy. These results support our hypothesis that delayed generalization in grokking is associated with the discovery of optimal structures.

## K    VISUALIZATION OF WEIGHTS AND GROKKING TICKET

To reveal the characteristics of the grokking ticket, we visualized the weight matrices and the corresponding masks of the grokking ticket, as shown in Figure 20. The task under consideration is Modular Addition, implemented using the following MLP architecture:

$$\text{softmax}(\sigma((\boldsymbol{E}_a + \boldsymbol{E}_b)W_{\text{in}})W_{\text{out}}W_{\text{unemb}}), \tag{2}$$

where $\boldsymbol{E}_a$ and $\boldsymbol{E}_b$ are the input embeddings, $W_{\text{in}}$, $W_{\text{out}}$, and $W_{\text{unemb}}$ are the respective weight matrices, and $\sigma$ denotes the activation function. This architecture models the relationship between inputs in the Modular Addition task.

Figure 20 visualizes the learned weight matrices $W_E$, $W_{\text{inproj}}$, $W_{\text{outproj}}$, and $W_U$ (top row) after generalization, as well as the corresponding masks from the grokking ticket (bottom row). The weight matrices exhibit periodic patterns, reflecting good structure learned during training. Furthermore, the grokking ticket masks align with these periodic characteristics, indicating that the grokking ticket has successfully acquired structures that are beneficial for the task of Modular Addition. These results highlight the ability of the grokking ticket to uncover meaningful patterns that contribute to the model's performance.

For comparison, Figure 21 show visualization of masks (structures) obtained by pruning-at-initialization (PaI) methods: Random, GraSP, SNIP, and SynFlow. Unlike the results shown in Figure 20, these methods do *not* exhibit periodic structures. This comparison highlights the superiority of the grokking ticket in acquiring structures that are more conducive to the Modular Addition task, further emphasizing its advantage over traditional PaI methods.

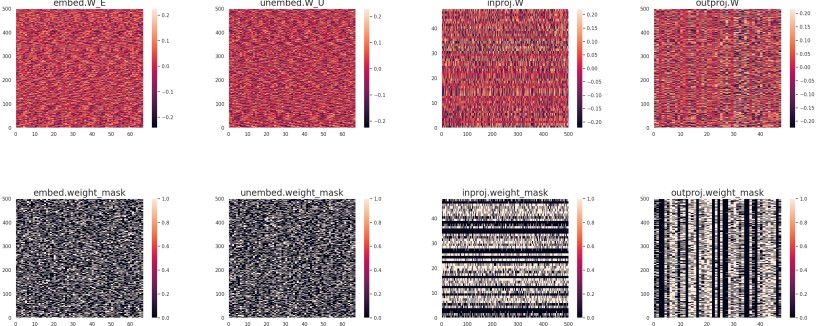

Figure 20: Visualization of weight matrices $W_E$, $W_{\text{inproj}}$, $W_{\text{outproj}}$, and $W_U$ (top), as well as the corresponding masks from the grokking ticket (bottom). Periodic patterns are observed in the weight matrices, and the masks of the grokking ticket reflect these characteristics. This indicates that the grokking ticket has acquired structures beneficial for the task (Modular Addition).

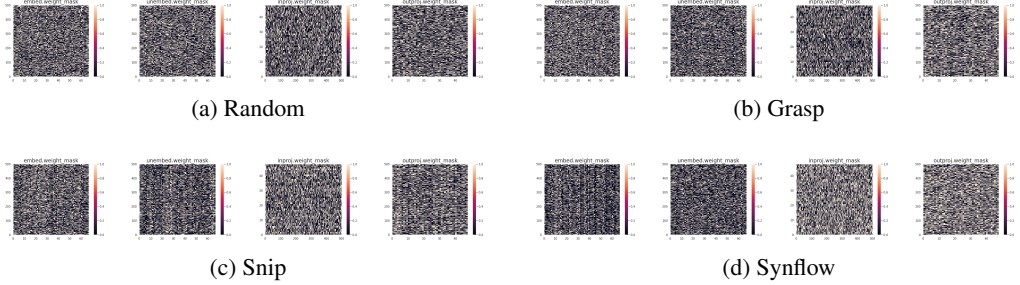

(a) Random    (b) Grasp

(c) Snip    (d) Synflow

Figure 21: Visualization of masks (structures) obtained by pruning-at-initialization (PaI) methods: Random, GraSP, SNIP, and SynFlow. Compared to grokking ticket in Figure 20, it can be observed that periodic structures are *not* achieved.

## L  SIGNIFICANCE OF PERIODICITY IN THE MODULAR ADDITION TASK

The modular addition task, defined as predicting $c \equiv (a + b) \mod p$, inherently involves periodicity due to the modular arithmetic structure. Nanda et al. (2023) demonstrated that transformers trained on modular addition tasks rely on periodic representations to achieve generalization. Specifically, the networks embed inputs $a$ and $b$ into a Fourier basis, encoding them as sine and cosine components of key frequencies $w_k = \frac{2k\pi}{p}$ for some $k \in \mathbb{N}$. These periodic representations are then combined using trigonometric identities within the network layers to compute the modular sum.

**Mechanism of Periodic Representations**

Nanda et al. (2023) reverse-engineered the weights and activations of a one-layer transformer trained on this task and found that the model computes:

$$\cos(w_k(a + b)) = \cos(w_k a)\cos(w_k b) - \sin(w_k a)\sin(w_k b),$$

$$\sin(w_k(a + b)) = \sin(w_k a)\cos(w_k b) + \cos(w_k a)\sin(w_k b),$$

using the embedding matrix and the attention and MLP layers. The logits for each possible output $c$ are then computed by projecting these values using:

$$\cos(w_k(a + b - c)) = \cos(w_k(a + b))\cos(w_k c) + \sin(w_k(a + b))\sin(w_k c).$$

This approach ensures that the network's output logits exhibit constructive interference at $c \equiv (a + b) \mod p$, while destructive interference suppresses other incorrect values.

Given this mechanism, it can be inferred that the model internally utilizes **periodicity**, such as the addition formulas, to perform modular arithmetic.

# M    HYPERPARAMETER EFFECTS ON $\tau_{grok}$

Figure 23 shows the effects of learning rate and weight decay on $\tau_{grok}$, defined in Section 3 as $t_{\text{train}}/t_{\text{test}}$.

**Learning Rate (Left):** With weight decay fixed at 1, $\tau_{grok}$ is NaN for lr=0.001 due to a failure to generalize. Larger learning rates (e.g., 0.01) lead to faster grokking, while smaller rates (e.g., 0.0001) significantly slow down the process.

**Weight Decay (Right):** With lr=0.01, smaller weight decay (e.g., 0.1) delays grokking, while larger values (e.g., 10) accelerates grokking, reducing $\tau_{grok}$.

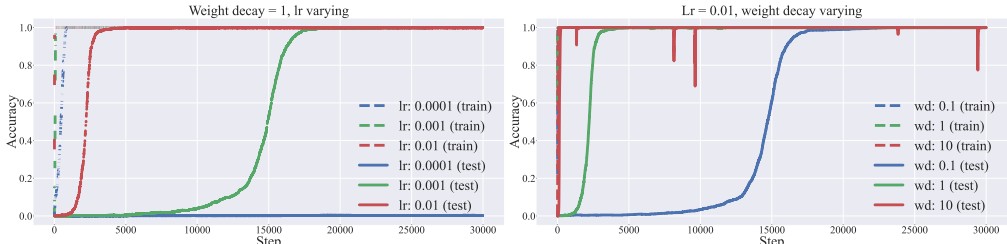

Figure 22: The effect of learning rate (left) and weight decay (right) on training and testing accuracy. **(Left)** Training and testing accuracy curves are plotted for different learning rates (lr) with a fixed weight decay of 1. A higher learning rate (e.g., 0.01) accelerates convergence but may cause instability, while lower learning rates (e.g., 0.0001) converge more smoothly but slower. **(Right)** Training and testing accuracy curves for different weight decay values (wd) with a fixed learning rate of 0.01. Larger weight decay values (e.g., 10) improve generalization, whereas smaller values (e.g., 0.1) result in delay.

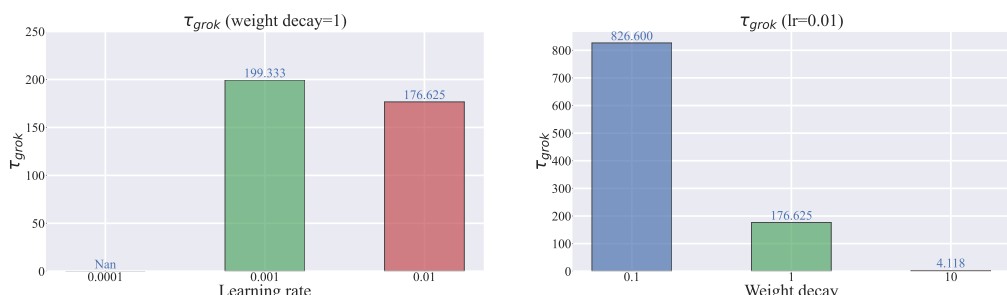

Figure 23: Comparison of $\tau_{grok}$ values under varying learning rates (left) and weight decay (right). **(Left)** $\tau_{grok}$ values are shown for different learning rates (0.0001, 0.001, and 0.01) with a fixed weight decay of 1. The learning rate of 0.001 fails to generalize, resulting in NaN values for $\tau_{grok}$. The results indicate that larger learning rates (e.g., 0.01) achieve grokking faster compared to smaller learning rates (e.g., 0.001). **(Right)** $\tau_{grok}$ values for varying weight decay (0.1, 1, and 10) with a fixed learning rate of 0.01. Lower weight decay values (e.g., 0.1) significantly slow down grokking, whereas larger weight decay values (e.g., 10) accelerates grokking.

