# OpenReview forum: "Bridging Lottery Ticket and Grokking: Understanding Grokking from Inner Structure of Networks"
_ICLR.cc/2025/Conference — Submitted to ICLR 2025_

### Official Review · Reviewer_2Wju · 2024-10-30

**Soundness:** 2
**Presentation:** 3
**Contribution:** 2
**Rating:** 5
**Confidence:** 4

**Summary:**

The paper found that given an appropriate lottery ticket of a specific task, the model generalizes much faster than the base model. They also found that the structure of the subnetwork changes rapidly during phase transition and pruning during training can accelerate generalization.

**Strengths:**

- The paper is well-written and easy to follow. It studies the hidden mechanism of grokking, which is an important question.

- The idea that a small proportion of generalized network is sufficiently for neural networks to generalize much faster and accelerate grokking is interesting.

**Weaknesses:**

- The hyperparameters (e.g. learning rate and weight decay) are not tuned. For example, I think for the modular addition task, the base model can generalize much faster than 25k steps given proper learning rate and weight decay. It is thus not fair to say the lottery ticket can accelerate 65 times in modular addition. Also the results in Table 1 would be more convincing if the hyperparameters are tuned.

- Why the network structure can be measured by Jaccard distance? It seems intuitive that when the model undergoes a phase transition (from memorization to generalization), its weight norm will change rapidly, causing the Jaccard distance to also increase rapidly.

**Questions:**

- Why the Jaccard distance can be used as a progress measure? The dynamics of Jaccard distance and that of test accuracy change show similar trend.

- Is it possible to obtain a good lottery ticket during memorization phase? It would be great if a mask can be found before the phase transition, so that it can be applied to the model and accelerates grokking.

---

> ### Author Response · Authors · 2024-11-23
> **Author Response (1/2)**
>
> We thank the reviewer for the constructive feedback. Please let us know if our responses in the following address your concerns.
>
> We revised the paper based on the reviewers’ comments, and the major edit was highlighted with coloring (purple). Please also check the updated manuscript.
>
> **W1**
> > The hyperparameters (e.g. learning rate and weight decay) are not tuned.
>
> We acknowledge the variability of grokking speed ($\tau_{\mathrm{grok}} = t_{\mathrm{train}} / t_{\mathrm{test}}$) with hyperparameter settings such as learning rate and weight decay.
> To clarify this, **Appendix M** includes experiments that illustrate how $\tau_{\mathrm{grok}}$ changes under various configurations. As you noted, hyperparameter tuning can influence the observed speed of grokking. However, our primary focus is **not** on the absolute speed of grokking but on the structural insights gained through the use of lottery tickets.
> Specifically, our study aims to explain how lottery tickets reduce delayed generalization from a structural perspective, as detailed in **Sections 4 and 5.** The findings in Section 3 serve as preparatory groundwork.
>
> Thus, while speed changes are observed, they are not the central message of our work. To clarify this emphasis, we revised our terminology throughout the paper, including in the **Abstract, Introduction, and Section 3**, replacing statements like “65 times faster” with “reduce delayed generalization.”
>
> This better reflects the essence of our findings and avoids overemphasizing specific speed improvements. We hope these clarifications address your concerns and provide a clearer context for our results.
>
>
> **W2 & Q1**
> > Why the network structure can be measured by Jaccard distance? It seems intuitive that when the model undergoes a phase transition (from memorization to generalization), its weight norm will change rapidly, causing the Jaccard distance to also increase rapidly.
>
> > Why the Jaccard distance can be used as a progress measure? The dynamics of Jaccard distance and that of test accuracy change show similar trend.
>
> While it is true that the weight norm decreases during the phase transition (as shown in  **Figure 6**), we explain in **Section 4.1** that weight norm alone cannot fully account for the phenomenon. This motivates the need for a new progress measure that better captures the underlying dynamics of structural changes during grokking. To address this, we propose the Jaccard distance as a novel progress measure.
>
> The Jaccard distance directly quantifies the distance between the structures of two networks, providing insights into the evolution of the inner structure as it transitions from memorization to generalization. Unlike the weight norm, which only reflects a global property of the weights, the Jaccard distance captures structural changes more explicitly, as evidenced by its alignment with test accuracy trends.

---

> > ### Author Response · Authors · 2024-11-23
> > **Author Response (2/2)**
> >
> > **Q2**
> > > Is it possible to obtain a good lottery ticket during memorization phase? It would be great if a mask can be found before the phase transition, so that it can be applied to the model and accelerates grokking.
> >
> >
> > In **Figure 4-(a)**, we show the performance of lottery tickets (non-grokking tickets) obtained during the memorization/transition phase before the model reaches generalization. The results indicate that the closer the lottery ticket is to a generalizing structure, the better base model performs. However, during the memorization phase, simply applying magnitude pruning to obtain a lottery ticket does not yield a good structure that facilitates generalization.
> >
> > On the other hand, in **Table 1**, we used the edge-popup algorithm to learn a mask (i.e., a structure) directly from the memorization phase. This algorithm updates only the mask without modifying the weights. The results show the model can transition from a memorization solution to a generalizing solution.
> >
> > Our findings suggest that while naive magnitude pruning during the memorization phase fails to produce a good lottery ticket, methods like edge-popup can effectively uncover a structure that accelerates grokking, even starting from the memorization phase.

---

> ### Author Response · Authors · 2024-11-25
> **A Reminder to Reviewer 2Wju**
>
> Thank you again for your valuable feedback on our paper. As we have not yet received a response, we would like to kindly remind Reviewer 2Wju to review these revisions.
>
> We would greatly appreciate it if you could consider whether our responses adequately address your comments.

---

> > ### Author Response · Authors · 2024-11-29
> > **Gentle Reminder to Reviewer 2Wju**
> >
> > Thank you again for your valuable feedback on our paper.
> >
> > To address your concerns, we have conducted experiments with various hyperparameters in **Appendix M** and updated the wording in the **Abstract, Introduction, and Section 3**.
> >
> > We would greatly appreciate it if you could review these updates and kindly reconsider your score in light of the revisions we have made to address your comments.

---

### Official Review · Reviewer_GMh9 · 2024-11-02

**Soundness:** 2
**Presentation:** 3
**Contribution:** 2
**Rating:** 5
**Confidence:** 4

**Summary:**

This paper studies the grokking phenomena of the subnetworks identified by the pruning methods in the Lottery Ticket Hypothesis. The findings of the paper are mainly three-fold. First, they show that, compared with the whole network, the subnetworks exhibits a shorter "grokking period", leading to a better generalization result faster. Second, they provide evidence to support that neither the weight norm nor the sparsity of the subnetwork is the crucial factor for the shorter "grokking period". Lastly, they show that the difference between the pruned subnetworks is related to the test accuracy at the point they are pruned, and that the subnetworks learns better representations from data.

**Strengths:**

1. The paper has a clear presentation of the experimental setup, which follows the previous works on grokking. This experimental design choice increases the effectiveness of their results.

2. In Section 3, the paper presented an interesting relationship between the pruning ratio and the duration of the grokking period.

3. In Section 5, the paper explored the connection between the test accuracy during training and the change of the masks.

**Weaknesses:**

1. The paper does not have a clear and unified argument. It seems that, instead of using the LTH as a tool to understand the grokking behavior of neural network training, the paper focus more on the grokking behavior of the pruned subnetworks. From this perspective, though they consist of interesting findings, the results of the paper lacks practical implications: we are not sure how the grokking behavior of the subnetworks is going to shed light on the grokking behavior of neural network in general from an "inner structure" perspective.

2. The paper does not decouple the behavior of "shorter grokking period" from the "easier training" property of the winning tickets. It was observed in the line of the LTH works that subnetworks that are winning tickets are easier to train, which include the behavior of achieving a better generalization result from a smaller number of epochs. It is not sure what is the de-facto distinction between this "easy-to-training" behavior and the result in this paper.

3. Some experimental design in Section 4 is weird. In particular, in Section 4.1 when constructing the neural network with the same weight norm as the pruned network, the paper directly shrinks each weight by a factor. There is no evidence that this method of reducing the weight norm will preserve the network performance or lead to better generalization performance, as is the case in the weight-decay training. Therefore, the paper could be comparing the lottery tickets with some arbitrarily bad neural network. Moreover, in Section 4.2, the paper compared the lottery tickets with PAI method, but it is known that PAI method usually sacrifices performance for better computation efficiency. Based on these experimental design, what Section 4 is doing is simply comparing lottery tickets with not-as-good neural network that has the same weight norms or sparsity, which makes the argument that the paper is trying to make trivial.

4. In Section 5.3, the paper tries to argue that the subnetworks has shorter grokking period because they learn better representations. Given that grokking is determined by generalization performance, it seems that, for the specific task design, the only way for the neural network to achieve good generalization is to learn the good representation (in other words, no benign overfitting is possible). This may not be the case for more complicated tasks.

**Questions:**

Which figure is line 486 trying to refer to? Right now it seems that there is a missing reference.

---

> ### Author Response · Authors · 2024-11-23
> **Author Response (1/2)**
>
> We thank the reviewer for the constructive feedback. Please let us know if our responses in the following address your concerns.
>
> We revised the paper based on the reviewers’ comments, and the major edit was highlighted with coloring (purple). Please also check the updated manuscript.
>
> **W1**
> > It seems that, instead of using the LTH as a tool to understand the grokking behavior of neural network training, the paper focus more on the grokking behavior of the pruned subnetworks. From this perspective, though they consist of interesting findings, the results of the paper lacks practical implications: we are not sure how the grokking behavior of the subnetworks is going to shed light on the grokking behavior of neural network in general from an "inner structure" perspective.
>
> We would like to clarify that our paper does not solely focus on pruned subnetworks. Instead, we build upon the concept of pruned subnetworks to derive insights about grokking behavior and extend the analysis to explore the characteristics of these subnetworks in depth.
> Specifically:
>
> 1. **Dynamics of Structure Acquisition (Section 5.1, Figure 7):**
>    Using the Jaccard distance as a metric for structural change, we analyze how the subnetwork evolves during grokking. This goes beyond simply observing pruned subnetworks and provides a dynamic perspective on how subnetworks adapt and contribute to grokking over time.
>
> 2. **Regularization Beyond Weight Decay (Section 5.2, Table 1):**
> From the observation that good structures are crucial for generalization, we introduce the edge-popup algorithm to directly explore structural optimization during training. By combining this structural exploration with traditional weight decay regularization, we demonstrate further improvements in generalization speed. These findings offer practical implications for designing new regularization strategies for grokking, extending beyond the conventional L2-based weight decay.
>
> In summary, our paper leverages pruned subnetworks not as the sole focus but as a foundation to uncover broader principles of grokking behavior. These findings provide practical implications, including a novel perspective on regularization and structural adaptation, which extend beyond the context of pruned subnetworks alone.
>
> We hope this addresses your concern. Please let us know if further clarification is needed.
>
> **W2**
> > The paper does not decouple the behavior of "shorter grokking period" from the "easier training" property of the winning tickets.
>
>
> We agree that winning tickets often exhibit "easy-to-train" properties, such as achieving generalization in fewer epochs, and we acknowledge that this observation itself is not novel. Our paper builds upon this known behavior by specifically connecting it to the grokking phenomenon, which introduces a unique perspective and deeper implications.
>
> For instance, as discussed in **Section 5.1 **, we observe abrupt changes in Jaccard distance during grokking, indicating structural evolution in the subnetworks. Furthermore, **Figure 8 ** reveal that these grokking tickets acquire task-specific structures, such as periodicity in Modular Addition, which are critical for solving the task and are directly tied to the grokking process.
>
> Thus, while faster generalization is a known property of winning tickets, our paper goes beyond this by examining how the discovery of task-specific good structures underpins the grokking phenomenon.
>
> To clarify these points, we have expanded the discussion in **Section 6**.
>
> We hope this clarifies our contribution, and we are happy to elaborate further if needed.

---

> > ### Author Response · Authors · 2024-11-23
> > **Author Response (2/2)**
> >
> > **W3**
> > > Based on these experimental design, what Section 4 is doing is simply comparing lottery tickets with not-as-good neural network that has the same weight norms or sparsity, which makes the argument that the paper is trying to make trivia
> >
> > The experiments in **Section 4** were specifically designed to test the validity of claims made in prior work by comparing our hypothesis against plausible alternative explanations. These comparisons are not intended to identify optimal configurations but rather to serve as counterfactual baselines to support our hypothesis.
> >
> > In **Section 4.1**, prior work suggests that grokking occurs when the weight norm transitions from an "overfitting zone" (large initial weight norm) to a "generalization zone" (smaller weight norm). According to this hypothesis, if weight norm alone were the determining factor for generalization, initializing the network in the "generalization zone" should eliminate delayed generalization altogether. By shrinking the weight norm of the network at initialization, we tested this hypothesis and found that delayed generalization still occurs. This result suggests that weight norm alone is insufficient to explain grokking, supporting our claim that good structures are more critical for grokking than weight norm.
> >
> > In **Section 4.2**, prior work also proposes that sparsity itself is a key factor in grokking. According to this view, if a sparse solution were provided at initialization, the model should generalize immediately. By comparing lottery tickets with a sparse initialization (via the PAI method), we demonstrated that sparsity alone does not lead to generalization. In our experiments, PAI methods do not achieve high performance. Instead, we use it to test the hypothesis that sparsity alone is sufficient to explain grokking.
> >
> > The observed results further support our hypothesis that **the specific structure of the subnetwork (i.e., the grokking ticket) plays a pivotal role in generalization**.
> >
> > In summary, the experimental designs in **Section 4** are carefully chosen counterfactuals based on the claims of prior work. They demonstrate that neither weight norm reduction nor sparsity alone can explain grokking, highlighting the importance of discovering good structures during training.
> >
> > We hope this addresses your concern, and we are happy to provide further clarifications if needed.
> >
> >
> > **W4**
> > > the paper tries to argue that the subnetworks has shorter grokking period because they learn better representations. Given that grokking is determined by generalization performance, it seems that, for the specific task design, the only way for the neural network to achieve good generalization is to learn the good representation．
> >
> > Our intention was to highlight that the relationship between good representations and sparsity, often discussed independently in prior studies, should be considered two sides of the same coin. In this paper, we aimed to bridge the findings from prior work emphasizing sparsity with the concept of good representations, uniting them under the notion of a "good structure."
> >
> > While it is indeed true that achieving good generalization necessitates the acquisition of good representations, we argue that it is not trivial for the network to acquire such representations as part of its structural organization.
> >
> > This perspective extends beyond the mere outcome of generalization, focusing on the structural and representational mechanisms that make this possible.
> >
> >
> > **Q1**
> > > Which figure is line 486 trying to refer to? Right now it seems that there is a missing reference.
> >
> > Thank you for pointing this out. This was a referencing error. In the revised version, we have corrected it and highlighted the change in purple.

---

> > > ### Comment · Reviewer_GMh9 · 2024-11-26
> > >
> > > Thank you so much for your response. Given the author's response, I still believe that the experiment design in Section 4 and 5 to be weird. Therefore, I will keep my score.

---

> ### Author Response · Authors · 2024-11-25
> **A Reminder to Reviewer GMh9**
>
> Thank you again for your valuable feedback on our paper. As we have not yet received a response, we would like to kindly remind Reviewer GMh9 to review these revisions.
>
> We would greatly appreciate it if you could consider whether our responses adequately address your comments.

---

### Official Review · Reviewer_CN4o · 2024-11-04

**Soundness:** 3
**Presentation:** 2
**Contribution:** 2
**Rating:** 6
**Confidence:** 3

**Summary:**

This work explores the phenomenon of grokking in neural networks, where models initially memorize training data without generalizing well, but after extended training, they suddenly begin to generalize effectively. Specifically, it investigates the relationship between grokking and the lottery ticket hypothesis---within a large neural network, there exist smaller, trainable subnetworks (or "winning tickets") that can achieve comparable performance to the original network. The authors introduce the concept of "grokking tickets", which are subnetworks identified during the generalization phase of training. They further show that the change of inner structure (subnetworks obtained by the magnitude pruning) highly correlates with the change in test accuracy.

**Strengths:**

The authors establish an innovative link between two significant phenomena in deep learning: grokking and the lottery ticket hypothesis, which provides new insights into understanding the generalization of neural networks. The experiments span multiple tasks, and the authors have also conducted thorough ablation studies to verify their hypothesis.

**Weaknesses:**

While this work presents an interesting investigation connecting grokking with the lottery ticket hypothesis, it is still unclear how the finding can be of practical usage, and some claims in this work still need more justifications (e.g., "the acquired good structure is linked to good representations") In particular, although the authors introduce Fourier Entropy to quantify the periodicity in the learned representations, it is still unclear how specific/different "periodicity" can be regarded as good representations. Also, the paper mainly explores grokking within fully connected networks and lacks a broad examination across various network architectures. For example, adding experiments across a wider range of architectures (e.g., Transformers) would increase the generalizability and appeal of the findings.

**Questions:**

1. How are grokking tickets fundamentally different from typical winning tickets found in lottery ticket hypothesis studies? Are there specific characteristics or properties that distinguish grokking tickets, or could they potentially overlap?
2. Do the authors have insights into the specific structures of the subnetworks (grokking tickets) that facilitate the transition from memorization to generalization? For example, do different types of tasks have different structures in their grokking tickets?
3. How does the finding generalize to other neural network structures (e.g., Transformer) and language-based tasks?
4. How does pruning affect the abilities of pretrained models in terms of grokking?
5. Why does periodicity in representations correlate with good performance?

---

> ### Author Response · Authors · 2024-11-23
> **Author Response (1/2)**
>
> We thank the reviewer for the constructive feedback. Please let us know if our responses in the following address your concerns.
>
> We revised the paper based on the reviewers’ comments, and the major edit was highlighted with coloring (purple). Please also check the updated manuscript.
>
> **W1 & Q5**
> >  it is still unclear how specific/different "periodicity" can be regarded as good representations.
>
> > Why does periodicity in representations correlate with good performance?
>
> We have addressed this in **Appendix L**, referencing (Nanda et al.)[1], demonstrating periodicity's significance in modular addition tasks. Below is a brief summary:
>
> The task, defined as predicting $c \equiv (a + b) \mod p$, inherently involves periodicity due to its modular arithmetic structure. Periodic representations align with this structure by encoding inputs $a$ and $b$ into a Fourier basis as sine and cosine components of key frequencies $w_k = \frac{2k\pi}{p}$.
>
> Using trigonometric identities, the model computes:
> $$
> \cos(w_k(a+b)) = \cos(w_k a) \cos(w_k b) - \sin(w_k a) \sin(w_k b),
> $$
> and logits are derived to ensure constructive interference at $c \equiv (a + b) \mod p$ :
> $$
> \cos(w_k(a+b-c)) = \cos(w_k(a+b))\cos(w_k c) + \sin(w_k(a+b))\sin(w_k c).
> $$
>
> This mechanism enables the model to generalize effectively by leveraging the modular arithmetic structure. We will also refer to this in **Section 5.3** of the main text for further clarity.
>
> If anything remains unclear, we are happy to provide further clarification.
>
> [1] Nanda et al., Progress measures for grokking via mechanistic interpretability, https://arxiv.org/abs/2301.05217
>
> **W2 & Q3**
> > the paper mainly explores grokking within fully connected networks and lacks a broad examination across various network architectures. For example, adding experiments across a wider range of architectures (e.g., Transformers) would increase the generalizability and appeal of the findings.
>
> > How does the finding generalize to other neural network structures (e.g., Transformer) and language-based tasks?
>
> We would like to clarify that our paper includes experiments with Transformers as well as MLPs. Specifically:
>
> 1. **Figure 2-(b)** in the main text presents results from Transformer-based experiments. These results demonstrate that, similar to MLPs, grokking tickets result in less delay of generalization compared to the base model in Transformers.
>
> 2. Additionally, we provide results from an NLP task (text sentiment analysis) using an LSTM architecture in **Appendix J**. In this task, we observed test accuracy improving almost simultaneously with training accuracy from the very beginning of the optimization process.
>
> Across a variety of tasks and architectures, including **Modular Arithmetic, MNIST, regression, sparse parity, and an NLP task**, our findings consistently show that having a good structure explains grokking. These tasks span architectures such as **MLPs, Transformers, and LSTMs**, supporting the generalizability of our claims.
>
> We hope this clarifies the breadth of our experiments and how they extend across diverse tasks and architectures. Please let us know if further clarification is needed.

---

> > ### Author Response · Authors · 2024-11-23
> > **Author Response (2/2)**
> >
> > **Q1 & Q2**
> > > How are grokking tickets fundamentally different from typical winning tickets found in lottery ticket hypothesis studies? Are there specific characteristics or properties that distinguish grokking tickets, or could they potentially overlap?
> >
> > > Do the authors have insights into the specific structures of the subnetworks (grokking tickets) that facilitate the transition from memorization to generalization? For example, do different types of tasks have different structures in their grokking tickets?
> >
> >
> > While the method for identifying grokking tickets is similar to that used in typical lottery ticket hypothesis studies, our work highlights distinct characteristics and unique insights specific to grokking:
> >
> > 1. **Fundamental Differences from Typical Lottery Tickets**:
> >    While typical lottery ticket studies emphasize the presence of subnetworks responsible for generalization, grokking tickets shed light on **dynamic structural changes during training** that are tightly linked to the transition from memorization to generalization. Specifically, as shown in **Figure 7**, we observe that grokking tickets emerge in tandem with improvements in test accuracy. This highlights a dynamic process of **structure acquisition** that is unique to grokking and not the focus of traditional lottery ticket studies.
> >
> > 2. **Task-Specific Structural Adaptations**:
> >    Grokking tickets exhibit characteristics that are highly tailored to the task at hand. For example, as shown in **Figure 8 (bottom)**, grokking tickets for Modular Addition display periodic structures, which are well-suited to solving this task. These task-specific structural features distinguish grokking tickets from typical lottery tickets, which are often analyzed more generally for their performance rather than their structural alignment with specific tasks. Additionally, visualizations of weight matrices and grokking tickets mask in **Appendix K** further illustrate that grokking tickets display periodicity and other unique characteristics not seen in traditional lottery tickets.
> >
> > These findings suggest that grokking tickets are more than just subnetworks that generalize well—they are task-specific structures optimized for generalization in the grokking context. This deeper structural perspective advances our understanding of how and why grokking occurs.
> >
> >
> >
> > **Q4**
> > > How does pruning affect the abilities of pretrained models in terms of grokking?
> >
> > Thank you for your insightful question. The phenomenon of grokking arises due to a distributional gap between the training and test data. During training, models initially overfit to the training data and later generalize to the test distribution.
> >
> > However, in the case of pretrained models like LLMs, grokking is less likely to occur because such models are trained on vast datasets that often already cover both training and test distributions. As a result, pretrained models typically do not exhibit the delayed generalization characteristic of grokking when evaluated on valid data within the scope of pretraining.
> >
> > Nevertheless, during fine-tuning (FT), pretrained models may encounter tasks where the fine-tuning data distribution differs significantly from the pretraining data distribution (e.g., answering in a specific QA format). In these scenarios, leveraging good structures, such as the proposed **grokking tickets**, could potentially lead to faster and more efficient learning. This is especially beneficial when fine-tuning on limited data, where structural guidance can accelerate generalization.

---

> > > ### Author Response · Authors · 2024-11-25
> > > **A Reminder to Reviewer CN4o**
> > >
> > > Thank you very much for your valuable feedback on our paper.  As we have not yet received a response, we would like to kindly remind Reviewer CN4o to review these revisions.
> > > To address your concerns, we have conducted experiments on new tasks and added further explanations about periodicity. These updates have been incorporated into the revised version of the paper.
> > >
> > > We would greatly appreciate it if you could consider whether our responses adequately address your comments.

---

> > > > ### Author Response · Authors · 2024-11-29
> > > > **Gentle Reminder to Reviewer CN4o**
> > > >
> > > > Thank you once again for your valuable review and constructive feedback on our paper.
> > > >
> > > > To reiterate, we have added experiments on a new task in **Appendix J** and provided further explanations about periodicity in **Appendix L** to address your concerns. These additions strengthen our claims.
> > > >
> > > > We would greatly appreciate it if you could review these updates and kindly reconsider your score in light of the revisions we have made to address your comments.

---

> > > > > ### Comment · Reviewer_CN4o · 2024-12-01
> > > > >
> > > > > Dear Authors,
> > > > >
> > > > > Thank you so much for your response! My concerns are cleared and I have raised my score to positive!

---

### Official Review · Reviewer_D6cU · 2024-11-07

**Soundness:** 2
**Presentation:** 3
**Contribution:** 2
**Rating:** 5
**Confidence:** 3

**Summary:**

This paper examines the phenomenon of "grokking," in which neural networks first achieve high training accuracy but poor generalization, then later switch to a generalization solution with more training. Earlier explanations explaining grokking along the lines of reducing weight norm are challenged, and it is argued that the identification of the "grokking tickets" specific subnetworks aligned with the lottery ticket hypothesis-play a key role in enabling generalization.

The authors provide empirical evidence that (1) grokking tickets can be used to overcome the phenomenon of delayed generalization observed in dense networks; (2) similar weight norms fail to overcome the need for long training if the subnetwork proper is missing, and finally, structure optimization alone--without weight updating--can transform memorization solutions into generalization solutions. These suggest that good sub-network search is more important to grokking than weight norm reduction, but it does offer a different perspective in regards to how generalization within neural networks happens.

**Strengths:**

Originality is demonstrated because the author tries to bridge two concepts that were seen independently before: the lottery ticket hypothesis and grokking. To put it in words, this can be shown through the proposition that the primary driving force behind grokking is not weight norm reduction but identification of a special subnetwork of "grokking tickets". Fresh angle of attack, challenging explanations of the form: what's really critical here is not just the simplicity of the model but the actual discovery of the subnetworks. One might expect this to have the effect of making a contribution that is more novel in its theoretical framing and prompts further exploration into the roles of sparsity and subnetwork structure more generally in generalization.

Quality-wise, the paper is sound because the experimental setup is robust, and careful comparisons across various models, including MLPs and transformers as well as tasks like modular arithmetic and MNIST, assure to critically analyze the idea. The methodology of the experiment controls very well, isolating the effects of subnetworks from confounding factors like weight norm. Indeed, that work can be labeled as comprehensive because it shows the use of multiple pruning techniques, establishment of a critical pruning ratio, and the use of experiments with edge-popup. All these conclusions strengthen the final conclusion drawn by the paper and reflect what should have been the actual efforts of the authors in being extremely cautious while testing their hypothesis.

The paper is clearly written in general. Authors described and explained the main concepts, including grokking tickets and their corresponding different pruning techniques used to find those subnetworks. Different experimental results of figures and graphs added clarity between grokking, weight norms, and subnetworks. Some parts of the theoretical sections, especially on the aspects that involve metrics, such as Jaccard similarity and frequency entropy, are highly supported by visual aids that enlighten the sub-network relationships and generalization.

The paper carries great value in terms of possibilities for reshaping discussions regarding delayed generalization and generalization mechanisms for neural networks. Treating the subnetworks as the focal point for grokking thus opens up avenues for research into model efficiency and interpretability, and maybe even efficiency in pruning. Moreover, if validated further, this approach is bound to have a considerable impact on thinking about training over-parameterized networks among practitioners and researchers and especially in identifying the optimal subnetwork. This approach for grokking could then be applied further to the realms of reinforcement learning and others, all pointing towards a wide-ranging impact of the discovery.

**Weaknesses:**

While extremely compelling and a new perspective on grokking, there are still some weaknesses of the paper, especially regarding theoretical justification, experimental scope, and clarity of interpretation.

One weakness lies in the theoretical motivation for why subnetworks (grokking tickets) are enough to explain grokking. Although experimentally subnetworks are shown to generalize faster than their dense counterparts, why it is that the subnetwork is generalizing and not in cases where weight norm fails remains slightly implicit. A discussion of some of the theoretical underpinnings of why some subnetworks well-perform, perhaps leveraging the insights from some recent sparsity studies on neural networks (e.g., Varma et al., 2023; Merrill et al., 2023) would significantly strengthen the conceptual framework and move it closer in line with prior work. Making such results more explicitly connectable to theoretical explanations of neural network generalization, such as in terms of double descent or simplicity bias, would go a long way in contextualizing the findings within a broader landscape of generalization theory.

The experimental setup is comprehensive but could be further extended to strengthen the generality of the results. For example, the main experiments are with simple modular arithmetic tasks and MNIST, which are suitable but quite simplistic and not necessarily representative of more complex distributions or task structures, as in natural language processing or computer vision. Further experiments would expand the complexity of the datasets and architectures used-for instance, BERT on NLP tasks or ResNet on CIFAR-10, which might be even leading towards more inference on whether the grokking tickets are consistently beneficial across a wide range of tasks and architectures. In this sense, an expanded effort would go to show how solid the results are and could further generalize their applicability.

Finally, even though the authors do account for the role of structural similarity in terms of Jaccard similarity in their grokking explanation, one might further explore what the implications of such a metric would be. Indeed, this could again relate to how variations in structural similarity over training align with the grokking process or consider how it would relate to properties of generalization. The other aspect where there is room for improvement includes the fact that the significance of subnetwork structures can be further ascertained through the use of other metrics for measuring the similarity of networks, based on weight sparsity patterns or neuron activations. Lastly, the paper could do much better in establishing the connection between experimental findings and practical implications.

For example, even though the study proves that a subnetwork can be learned without weight decay, it may be more helpful to know how this might impact the practices in the real world in terms of training or pruning neural networks. It would also provide more concrete recommendations or at least some hint on possible applications of grokking tickets to better connect the theoretical insights to practice. This will not only make the contribution much more vivid, but also emphasize how important the findings are for the more general community of practitioners of machine learning. In other words, the paper does provide a good foundation, yet it could further be improved with a more detailed theoretical justification, broader experimental validation across complex tasks, a deeper exploration of structural metrics, and clearer practical implications. Such refinements would make the work more comprehensive and more impactful in advancing our understanding of grokking and neural network generalization.

**Questions:**

1. It would be great if you provide more theoretical insight into why it is enough for generalization to have the presence of certain subnetworks (grokking tickets)? It would be interesting to know if there is an undercurrent mechanism, besides the empirical evidence, that could be given to support the idea that it was really the subnetworks that drive the transition from memorization to generalization and not the weight norm. Lastly, would it be possible to learn other relevant theories on generalization, for example, simplicity bias or double descent, which could place these findings into a wider context?

2. Although the paper does very nicely on modular arithmetic and MNIST, how would grokking tickets generalize to a few domains, perhaps such as NLP, like using a model such as BERT? Or image recognition, such as applying a CNN to CIFAR-10? This would help in generalizing the results. Can the authors comment on whether grokking tickets applies to other forms of tasks or known limitations for those other domains?

3. Since the results show generalization with the appropriate subnetworks instead of depending on weight decay, do the authors have any insights on practical takeaways? For instance, how do these grokking tickets that were discovered alter regimes of training, pruning of models, and the choice of architecture? Including more specific recommendations on how to leverage grokking tickets in practice better opens up findings for practitioners to put into action.

4. The paper introduces Jaccard similarity to measure structural similarity between subnetworks, yet does not delve enough into the implications of this metric. Could the authors go into additional detail about which variations in structural similarity correlate with stages of grokking? Specifically, does some degree of Jaccard similarity correspond to a "critical" sub-network structure predictive of successful generalization? Further research on the nature of structural similarity resulting from training might shed further light on grokking tickets.

5. The authors introduce a critical pruning ratio, such as 0.81, needed to achieve generalization without weight decay. Would the authors comment upon how that ratio could vary by architectures and datasets? A more in-depth examination of how the result is sensitive to this pruning rate and others can help to explain how reliably grokking tickets can be identified across different setups.

6. While the authors have employed Jaccard similarity and frequency entropy to compute quality metrics of subnetworks, do they also explore other metrics for validation of soundness of their results? For example, capturing the degree of similarity in activation of neurons or any such sparsity-based metrics might prove the significance of certain structures of subnetworks as well. This will likely reiterate that in fact subnetworks are prime essentials for generalization.

7. The authors show that typical pruning-at-initialization methods, such as SNIP and Synflow, cannot efficiently produce grokking tickets. Are the authors willing to provide more analysis into why such classical PaI methods do not elicit generalization? For example, is something inherently different between the identified subnetworks by the former methods compared to grokking tickets? This comparison would help clarify the unique properties of grokking tickets and guide future improvements in these techniques.

8. The experimental results seem to suggest that the structures of subnetworks change over time. Can the authors provide some illustrations-for example, weight heatmaps or connectivity graphs of subnetworks-probing the evolution through multiple training phases? Such illustrations would make the process more intelligible, where grokking tickets form and generalize, and serve to additionally demonstrate their role in delayed generalization.

9. The paper records that grokking tickets facilitate the generalisation when compared with dense networks. Is it possible for the authors to probe for a predictable point during the training regime where these sub-networks first arise? Investigating whether there is some measurable "onset" of generalization with grokking tickets, maybe utilising Jaccard similarity or other measures, might reveal important transition points in the training regime.

10. Some phenomena of generalization, like the double descent phenomenon, have been studied to some large extent. Would it be interesting if the authors were able to delineate how grokking tickets might relate to these other phenomena? For example, is an appearance of grokking tickets associated with first descent in a curve describing a situation of double descent? If one is able to draw these relations, it may give grokking tickets a context of being part of a bigger picture of generalization dynamics of neural networks.

---

> ### Author Response · Authors · 2024-11-23
> **Author Response (1/3)**
>
> We thank the reviewer for the constructive feedback. Please let us know if our responses in the following address your concerns.
>
> We revised the paper based on the reviewers’ comments, and the major edit was highlighted with coloring (purple). Please also check the updated manuscript.
>
> **W1 & Q1**
> > the theoretical motivation for why subnetworks (grokking tickets) are enough to explain grokking.
>
> > It would be great if you provide more theoretical insight into why it is enough for generalization to have the presence of certain subnetworks (grokking tickets)?
>
> While not fully grounded in theory, but it is evident that task-adaptive structures contribute significantly to generalization. For instance, as demonstrated in Neyshabur [1], which is based on the theory of Minimum Description Length (MDL), incorporating $\beta$-Lasso regularization into fully connected MLPs facilitates the emergence of locality—resembling the structures found in CNNs—leading to improved performance in image-related tasks. This insight aligns closely with the motivation of our work, as mentioned in the introduction of our paper: to analyze the delayed generalization in grokking from the perspective of network structure.
>
> While the relationship between good structures and generalization has been extensively studied in deep learning, the connection between grokking and network structural properties remains underexplored. Our research seeks to bridge this gap by investigating how specific subnetworks (grokking tickets) contribute to generalization.
>
> We hypothesize that the reason grokking tickets generalize well is their ability to acquire a task-adaptive structure, similar to how CNNs adapt to image data. This hypothesis is substantiated by the findings in **Section 5.3**, where we show that the subnetworks acquire periodicity—a critical characteristic for Modular Addition tasks. This periodic structure in Modular Addition tasks can be interpreted analogously to the local structures in image tasks captured by CNNs. These results provide insight into the importance of a network's inner structure for achieving generalization.
>
> To make this point clearer, we have revised the following sections of our paper:
>
> - **Section 5.3**: We have updated this section to make it clearer that grokking tickets acquire task-specific, beneficial structures. The revised text highlights how these structures contribute to generalization.
> - **Appendix K**: Related to Q7 and Q8, we have added the visualizations of weight matrices and masks of the grokking tickets after generalization. This addition provides a clearer illustration of how grokking tickets exhibit task-relevant structures (periodicity).
> - **Abstract and Introduction**: We have updated these sections to reflect the changes and emphasize our motivation and the key insights regarding the role of task-adaptive structures in grokking tickets.
>
>
> We believe these revisions address the reviewer's concerns and offer a clear explanation of why subnetworks (grokking tickets) are sufficient to account for the phenomenon of grokking.
>
> [1] Neyshabur, Towards Learning Convolutions from Scratch, https://arxiv.org/abs/2007.13657
>
> **Q8**
> > Can the authors provide some illustrations-for example, weight heatmaps or connectivity graphs of subnetworks-probing the evolution through multiple training phases?
>
> We have included visualizations of the weight matrices after generalization and the masks of the grokking ticket in **Figure 22** (**Appendix K**). In these results, periodic patterns are observed in the weight matrices, and the masks of the grokking ticket reflect these characteristics. This indicates that the grokking ticket has acquired structures that are beneficial for the task (Modular Addition).
>
> These visualizations additionally demonstrate the role of grokking tickets in delayed generalization through their ability to acquire periodic structures.
>
> **Q7**
> > Are the authors willing to provide more analysis into why such classical PaI methods do not elicit generalization?
>
> We have provided additional analysis in **Appendix K**, specifically in **Figure 23**, where we compare the masks (structures) obtained by pruning-at-initialization (PaI) methods such as Random, GraSP, SNIP, and SynFlow. Unlike the results of the grokking ticket shown in **Figure 22**, these PaI methods do **not** exhibit periodic structures.
>
> This comparison highlights the superiority of the grokking ticket in acquiring structures that are more conducive to the Modular Addition task, further emphasizing its advantage over traditional PaI methods.

---

> ### Author Response · Authors · 2024-11-23
> **Author Response (2/3)**
>
> **W2 & Q2**
> > The experimental setup is comprehensive but could be further extended to strengthen the generality of the results.
>
> > Although the paper does very nicely on modular arithmetic and MNIST, how would grokking tickets generalize to a few domains, perhaps such as NLP, like using a model such as BERT?
>
> To address the concern about the generality of grokking tickets across different domains, we have included an analysis in **Appendix J**, where we evaluated grokking tickets on a sentiment analysis task using the IMDb dataset (Maas et al., 2011)[2]. This dataset contains 50,000 movie reviews classified as positive or negative, processed with the 1,000 most frequent words and tokenized into arrays of indices. For classification, we employed a two-layer LSTM model, and the experimental details are provided in **Appendix J.**
>
> The results show the base model rapidly achieves 100% training accuracy; however, its test accuracy remains at 50%(chance rate) until approximately 10k optimization steps, after which it begins to improve. On the other hand, the grokking ticket demonstrates a different behavior, with test accuracy improving almost simultaneously with training accuracy from the very beginning of the optimization process. This suggests that the grokking ticket's structural properties facilitate faster and more efficient generalization.
>
> These findings extend the applicability of grokking tickets beyond modular arithmetic and MNIST to NLP tasks such as sentiment analysis. They also underscore the potential of grokking tickets to uncover meaningful patterns in different domains, further demonstrating their versatility and generality.
>
> [2] Maas et al., Learning Word Vectors for Sentiment Analysis, https://aclanthology.org/P11-1015/
>
> **W3 & Q4**
> >  the authors do account for the role of structural similarity in terms of Jaccard similarity in their grokking explanation, one might further explore what the implications of such a metric would be
>
> >  Could the authors go into additional detail about which variations in structural similarity correlate with stages of grokking?
>
> Our hypothesis is that during grokking, the network acquires a good structure that is beneficial for generalization. For the modular addition task, this corresponds to a periodic structure, as demonstrated in **Figure 8 (bottom).**
>
> The Jaccard similarity metric directly reflects how the network structure (as captured by the magnitude pruning mask) evolves during training. **Figure 7** shows that the structure changes abruptly, corresponding with the sharp increase in test accuracy, indicating a significant structural transformation of the network.
> When we delve deeper and combine these findings with the results on periodic structures (**Figure 8**), we observe that during grokking, the network rapidly transitions to a periodic structure—one that is highly suitable for the task—at the point when test accuracy improves dramatically.
>
>
> **W4 & Q3**
> > it may be more helpful to know how this might impact the practices in the real world in terms of training or pruning neural networks.
>
> > the results show generalization with the appropriate subnetworks instead of depending on weight decay, do the authors have any insights on practical takeaways?
>
>
> Our findings that highlight the importance of good subnetworks suggest a new perspective on regularization. While weight decay traditionally regularizes the L2 norm of weights, our results imply that effective regularization should focus on  **discovering good structures** within the network. In this sense, weight decay can be seen as a proxy for encouraging such structure discovery indirectly.
>
> For example, as shown in  **Table 1**, adding the edge-popup algorithm, which explicitly explores structure  (optimizes masks), to weight decay resulted in faster generalization. This demonstrates that integrating **structural exploration into training can enhance performance** beyond what weight decay alone achieves.
>
> These insights suggest that practitioners may improve generalization by incorporating methods that directly optimize beneficial structures rather than solely relying on traditional regularization techniques like weight decay. Our results pave the way for developing new, structure-oriented regularization techniques to better leverage the benefits of grokking tickets in practical applications.
>
> To make this point clearer, we have revised **Section 5.2** to elaborate on the implications of the edge-popup algorithm's results.

---

> > ### Author Response · Authors · 2024-11-23
> > **Author Response (3/3)**
> >
> > **Q6**
> > > While the authors have employed Jaccard similarity and frequency entropy to compute quality metrics of subnetworks, do they also explore other metrics for validation of soundness of their results?
> >
> > Thank you for your insightful suggestion. While we agree that additional metrics, such as neuron activation similarity or sparsity-based measures, could provide further validation, we believe that frequency entropy is already a robust metric for capturing the periodicity of subnetwork structures.
> >
> > To make our findings even clearer, we have also included visualizations of the weight matrices and grokking ticket masks in **Appendix K**. These visualizations offer an intuitive perspective on the structural patterns within subnetworks, complementing the quantitative results presented in the main paper.
> >
> > We appreciate your suggestion and believe that these visualizations further reinforce the importance of specific subnetwork structures for generalization. If you have additional metrics or approaches in mind, we would be happy to explore them.
> >
> >
> > **Q9**
> > > Is it possible for the authors to probe for a predictable point during the training regime where these sub-networks first arise?
> >
> > Thank you for your insightful question. We agree that identifying a predictable point during training where sub-networks first arise could provide valuable insights into the dynamics of generalization. As you suggested, Jaccard distance shows potential for predicting the onset of generalization.
> >
> > For example, in **Appendix C**, we provide results for the polynomial task and sparse parity task, showing the relationship between Jaccard distance and test accuracy during training. Specifically, in **Figure 14-(a)**, we observe a sharp increase in Jaccard distance preceding the rise in test accuracy. This suggests that the internal structure of the model begins to change before generalization is reflected in the test performance.
> >
> > These findings highlight the utility of Jaccard distance as a measure to probe for structural transitions that may predict generalization. We appreciate your suggestion and believe it opens up promising avenues for further exploration in understanding the dynamics of grokking tickets.
> >
> >
> > **Q10**
> > > Would it be interesting if the authors were able to delineate how grokking tickets might relate to these other phenomena?
> >
> > We agree that connecting grokking tickets to broader generalization phenomena, such as double descent, is an important direction. Our findings suggest that generalization in neural networks involves two distinct optimization processes: weight optimization and structural optimization. This perspective provides a potential interpretation of double descent dynamics.
> >
> > The first descent may correspond to weight optimization and the second descent to structural optimization. Supporting this, **Table 1** in our study demonstrates that structural optimization, achieved through edge-popup, is critical for generalization, aligning with the hypothesis that these two forms of optimization underpin double descent behavior.
> > However, it is important to note that grokking and double descent differ in their x-axes: grokking examines training steps, while double descent studies often use parameter count. Despite this distinction, our findings reveal that the structural changes associated with grokking tickets complement the dynamics observed in double descent, offering a unified perspective on generalization mechanisms in neural networks.

---

> > > ### Author Response · Authors · 2024-11-25
> > > **A Reminder to Reviewer D6cU**
> > >
> > > Thank you once again for your valuable review and constructive feedback on our paper. As we have not yet received a response, we would like to kindly remind Reviewer D6cU that we have thoroughly addressed your concerns. Specifically, we have conducted additional experiments on new tasks and enhanced our analysis, including an in-depth examination of weight matrices. These updates have been incorporated into the revised paper, which we believe has been significantly strengthened as a result.
> > >
> > > We would greatly appreciate it if you could review these updates and kindly reconsider your score in light of the revisions we have made to address your comments.

---

> > > > ### Comment · Reviewer_D6cU · 2024-11-25
> > > >
> > > > Thanks for the detailed feedback and additional experiments. Considering the changes you have made, I have altered my score. Please make sure to incorporate all these changes in the paper.

---

> > > > > ### Author Response · Authors · 2024-11-26
> > > > >
> > > > > Thank you for your response.
> > > > >
> > > > > We will work on addressing the points you have highlighted and incorporate them into the paper.
> > > > >
> > > > > We would like to express our sincere gratitude once again to the reviewers for their thoughtful feedback, which has greatly helped us refine the contributions, positioning, and limitations of our work.

---

### Author Response · Authors · 2024-11-23
**Summary of Revision in Author Response**

We appreciate detailed reading and suggestive feedback from all the reviewers. We revised the paper based on the reviewers’ comments, and **the major edit was highlighted with coloring (purple).**

The key changes are summarized below:
- Added experiments on grokking tickets for NLP tasks in **Appendix J**, expanding the scope of our research. (Reviewers D6cU, CN4o)
- Included visualizations of weight matrices and grokking ticket masks in **Appendix K**, making the specific structures of grokking tickets more comprehensible. (Reviewers D6cU, CN4o)
- Added visualizations of the PaI mask in **Appendix K**. (Reviewers D6cU)
- Expanded the explanation of the importance of periodicity in Modular Addition tasks in **Appendix L**. (Reviewer CN4o)
- Addressed the effects of hyperparameters (learning rate, weight decay) in **Appendix M**. (Reviewer 2Wju)
- Revised the description of "65 times faster" to "reduce delayed generalization" in the **abstract** and **Section 3**. (Reviewer 2Wju)
- Emphasized that grokking tickets possess unique periodic structures in the **introduction** and **Section 5.3**. (Reviewers D6cU, )
- Clarified the practical implications of pruning for improving generalization in **Section 5.2**. (Reviewers D6cU)
- Added a discussion on the differences between grokking tickets and lottery tickets in **Section 6** (Reviewer GMh9)
- Added related work on our motivation in the **Introduction**. (Reviewer D6cU)
- Corrected reference errors. (Reviewer GMh9)

We provide a detailed explanation of these and other minor revisions in our responses to the individual reviews below. Once again, we would like to express our gratitude to the reviewers for their insightful feedback, which we believe has significantly enhanced our paper.

---

### Meta-Review · Area_Chair_YoN9 · 2024-12-23

**Metareview:**

This paper studies the connection between lottery tickets and grokking. The authors make the observation that using lottery tickets chosen at the time of test generalization can mitigate grokking challenges. Here, grokking refers to the phenomena that, for certain tasks such as modular arithmetic, test accuracy is observed to reach high accuracy much later than training accuracy. Overall, the reviewers and the AC believe that this is a nice connection and authors make various observations around this.

On the other hand, the paper has some fundamental issues which results in the reject recommendation. One issue is the concept of "grokking tickets" and how it is different from LTH (this concern is shared by some reviewers as well). The definition of "grokking tickets" is solely based on choosing a lottery ticket based on its test accuracy. This definition is not really specific to grokking and this sounds like an obvious thing to do. I feel like naming is misleading essentially. I understand that this choice can particularly benefit tasks where grokking (train-test discrepancy) is observed but the definition itself is not grokking specific. Additionally, the LTH has a rich literature. I would be surprised if nobody has studied related criteria (based on test error) to decide how to pick the lottery tickets. The related work section (in Page 10) has only a short paragraph and a single citation on LTH which makes me concerned about potential missing related work here. A second issue is that, given this is an empirical work, the current set of experiments don't meet the bar in terms of how comprehensive they are (also shared by some reviewers). A final issue is lack of theoretical depth and limited methodological contributions.

**Additional Comments On Reviewer Discussion:**

During the rebuttal, reviewers raised concerns about the novelty of the proposed methods and insufficient theoretical grounding. Authors responded by emphasizing their experimental rigor and introducing new visualizations of grokking tickets. However, these additions failed to address the fundamental critiques, particularly the lack of clarity in distinguishing the grokking tickets' role from established pruning techniques and how they are fundamentally different from standard LTH. While some reviewers appreciated the practical insights, the overall consensus leaned toward rejection due to weak theoretical contributions and a not fully-complete empirical narrative.

---

### Decision · Program_Chairs · 2025-01-22

Reject